



# Assessing global water mass transfers from continents to oceans over the period 1948–2016

Denise Cáceres[1], Ben Marzeion[2], Jan Hendrik Malles[2], Benjamin Gutknecht[3], Hannes Müller Schmied[1,4], Petra Döll[1,4]

[1]Institute of Physical Geography, Goethe University Frankfurt, Frankfurt am Main, Germany
[2]Institute of Geography and MARUM, University of Bremen, Germany
[3]Institut für Planetare Geodäsie, Technische Universität Dresden, Germany
[4]Senckenberg Leibniz Biodiversity and Climate Research Centre Frankfurt (SBiK-F), Frankfurt am Main, Germany

*Correspondence to*: Denise Cáceres (d.caceres@em.uni-frankfurt.de)

**Abstract.** Continental water mass change affects ocean mass change (OMC). Assessing the net contribution, however, remains a challenge. We present an integrated version of the WaterGAP global hydrological model that is able to simulate total continental water storage anomalies (TWSA) over the global continental area (except Greenland and Antarctica) consistently by integrating the output from the global glacier model of Marzeion et al. (2012) as an input to WaterGAP. Monthly time series of global mean TWSA obtained with an ensemble of four variants of the integrated model, corresponding to different precipitation input and irrigation water use assumptions, were validated against an ensemble of four TWSA solutions based on GRACE satellite gravimetry over January 2003 to August 2016. The overall fit to GRACE, measured by the Nash–Sutcliffe efficiency (NSE) coefficient, was found to be 0.87. By decomposing the original TWSA signal into its seasonal, linear trend and inter-annual components, we find that the seasonal amplitude and phase are very well reproduced (NSE = 0.88), the linear trend is overestimated by 30–50% (NSE = 0.65) and inter-annual variability is captured to a certain extent (NSE = 0.57) by the integrated model. During the period 1948–2016, we find that continents lost 34–41 mm of sea level equivalent (SLE) to the oceans, with global glacier mass loss accounting for 81% of the cumulated mass loss and glacier-free land water storage anomalies (LWSA) accounting for the remaining 19%. Over 1948–2016, the mass gain on land from impoundment of water in man-made reservoirs, equivalent to 8 mm SLE, was offset by the mass loss from water abstractions, amounting to 15–21 mm SLE and reflecting a cumulated groundwater depletion of 13–19 mm SLE. Climate-driven LWSA are highly sensitive to precipitation input and correlate with El Niño Southern Oscillation multi-year modulations. Significant uncertainty remains in trends of modelled LWSA, which are highly sensitive to simulation of irrigation water use and man-made reservoirs.

## 1. Introduction

Global mean sea-level rise (SLR) in recent decades is mainly caused by anthropogenic climate change (Slangen et al., 2016), but is also affected by direct human interventions such as impoundment of water in man-made reservoirs and water abstractions (Church et al., 2013; Oppenheimer et al., 2019). Since the beginning of the altimetry era, several satellite missions have produced continuous measurements of sea-level height to monitor its evolution. These records have given rise





to multiple studies on global mean sea-level change and its different components (Gregory et al., 2013; Cazenave et al., 2014; Dieng et al., 2017). It is well-known that sea-level change can be decomposed into a steric component (i.e. thermal
expansion and salinity change) and a mass component (i.e. change of ocean mass). The Gravity Recovery and Climate Experiment (GRACE) joint mission of the National Aeronautics and Space Administration (NASA) and the German Aerospace Center has enabled, for the first time in history, the monitoring of spatially and temporally distributed ocean mass change (OMC). The GRACE twin satellites have produced records of the Earth's gravity field anomalies extending from April 2002 to August 2016. Time series of OMC can be derived from the gravity field observations, upon applying several
post-processing corrections to the raw data, e.g. regarding glacial isostatic adjustment and leakage from the continental mass signal.

According to the principle of water mass conservation in the Earth system, OMC can be decomposed into temporal changes in mass of 1) Greenland and Antarctica ice sheets, 2) glaciers, 3) water stored on the continents and 4) atmospheric water vapour. If the water mass in these four compartments decreases, ocean mass increases. Hereafter, we refer to water mass
changes of glaciers (different from ice sheet peripheral glaciers) as land glacier water storage anomaly (LGWSA) and the water mass changes on glacier-free land as land water storage anomaly (LWSA). LWSA is defined as all water stored in rivers, lakes, wetlands, man-made reservoirs, snow pack, canopy, soil and aquifers, excluding glaciers (Church et al., 2013; Scanlon et al., 2018). The term anomaly is used to express that the absolute mass of water in glaciers and on glacier-free land is not known and also not relevant for determining the contribution of these mass changes to OMC. The sum of LGWSA and
LWSA is equal to the total continental water storage anomaly (TWSA). It can be assumed that the majority of any mass decrease in TWSA leads to increased mass in the ocean in particular due to the small water storage capacity of the atmosphere (Dieng et al., 2015; Cazenave, 2018). TWSA and thus global water mass transfers from continents to oceans can be derived either from estimates of its two components or from GRACE observations over the continents.

LGWSA constitutes a major contribution to OMC. There are several global glacier models capable of simulating LGWSA at
global scale (Hirabayashi et al., 2013; Marzeion et al., 2012; Huss and Hock, 2015). Using such a model, Marzeion et al. (2015) estimated a global glacier mass loss corresponding to $63.2 \pm 7.9$ mm SLE between 1902 and 2005. Projections for the twenty-first century show that mass loss of glaciers will continue to be a significant contribution to OMC and thus SLR (Slangen et al., 2017; Hock et al., 2019). The contribution of LWSA to OMC remains highly uncertain (Cazenave, 2018). LWSA is not only driven by climate variability and change but also human activities; LWSA is thus the sum of climate-
driven LWSA ($LWSA_{clim}$) and human-driven LWSA ($LWSA_{hum}$).

Different approaches have been employed to estimate LWSA. Dieng et al. (2015) used a global ocean mass budget approach to quantify LWSA over 2003–2013; they compared GRACE-based OMC to the sum of mass components derived from independent products, except for LWSA, which was the unknown quantity to be estimated. In other studies, this component was extracted from GRACE-derived TWSA (Reager et al., 2016; Rietbroek et al., 2016). Another approach to estimate
LWSA is the use of global hydrological and land surface models (GHMs). GHMs are driven by time series of spatially





distributed climate variables such as precipitation and temperature that are usually derived from atmospheric reanalysis. They simulate water flows and water storages on continents, distinguishing water storage anomalies in individual water storage compartments. For example, the WaterGAP GHM simulates anomalies in snow, canopy, soil, groundwater and surface water bodies (SWB) including rivers, lakes, man-made reservoirs and wetlands (Döll et al., 2014). Furthermore,
some GHMs like WaterGAP and PCR-GLOBWB (Sutanudjaja et al., 2018) explicitly simulate the impact of human water use and of impoundment of water in man-made reservoirs on LWSA. Wada et al. (2017) assessed components of LWSA based on 1) modelling groundwater depletion with PCR-GLOBWB, 2) estimating impoundment behind dams by adding storage capacities of reservoirs, 3) assuming the $LWSA_{clim}$ estimate of Reager et al. (2016) and 4) very roughly estimating storage losses in endorheic lakes (Caspian Sea and Aral Sea), wetlands and due to deforestation based on literature. Most
GHMs do not simulate the effect of glacier mass change on hydrological flows and storages; areas that in reality are covered by glaciers (hereafter glacierized areas) are treated as normal (i.e. non-glacierized) areas. However, some exceptions exist; for instance, the Community Land Model version 5.0 (CLM5) includes a dynamic fractional glacier area and simulates surface mass balance (SMB) of glaciers all over the world (Lawrence et al., 2019).

In the last decades, many studies have compared water storage anomalies derived from GHMs and GRACE at different
temporal and spatial scales (Di Long et al., 2017; Scanlon et al., 2018; Scanlon et al., 2019). However, the globally averaged water storage anomaly on the continents simulated by GHMs, which is the relevant variable for the assessment of global mean SLR, has not yet been compared to GRACE observations. While comparison of GRACE TWSA to GHM LWSA (i.e. total water storage estimated without considering glaciers) is consistent for regions without significant glaciers, this is not the case for analyses of globally averaged continental water mass changes, in particular because the TWSA trend is strongly
impacted by LGWSA. In OMC assessments, the common practice is to add the trends of the two independently estimated components LWSA and LGWSA (Gregory et al., 2013; Dieng et al., 2017; Cazenave, 2018). This, however, has the inherent problem of double-counting mass changes in glacierized areas, which are accounted for by both glacier mass loss estimates and GHMs (as non-glacierized areas). It has not yet been assessed whether this double-counting is significant at the global scale. In addition, glacier mass variability affects the water flow dynamics on the continents as well as the seasonality of
water transfers from the continents to the oceans. This can only be taken into account by integrating glacier dynamics into GHMs.

One of the advantages of global models over GRACE observations for estimating global water mass transfers from continents to oceans is that models are not limited to the GRACE observational period, but can provide estimates for longer time periods. Furthermore, in the case of GHMs, another advantage is the possibility to disaggregate LWSA into the
contributions of its individual components. On the downside, global models are subject to many sources of uncertainty (e.g. input data, calibration approach, parameterization, model structure) and their global-scale validation is difficult mainly due to a lack of observations of e.g. streamflow, groundwater level or soil moisture. In this respect, satellite observations with global coverage like GRACE are of considerable interest for the hydrological community. Because of the spatial resolution





of GRACE (~300–500 km), it is known that the derived TWSA should not be used to validate models at smaller spatial
scales. However, considering that the quality of GRACE estimates increases with decreasing resolution (i.e. larger areas),
they can be of great use to constrain and improve global and regional hydrological models, as proved by many studies
(Güntner, 2008; Tangdamrongsub et al., 2018; Werth and Güntner, 2010).

We did a long-term (1948–2016) assessment of TWSA and thus the contribution of water transfers from continents (except
Antarctica and Greenland) to OMC. Our assessment aimed at quantifying this contribution during the period 1948–2016, as
well as identifying its main drivers and components. In a first instance, we disaggregated TWSA into the contributions of
LGWSA, LWSA$_{clim}$ and LWSA$_{hum}$. We further disaggregated LWSA$_{hum}$ by quantifying separately the effect of water
impoundment in reservoirs (LWSA$_{res}$) and the effect of water abstraction (LWSA$_{abs}$). TWSA estimates were obtained by
combining two state-of-the-art global models; the global glacier model GGM of Marzeion et al. (2012) and the GHM
WaterGAP (Döll et al., 2003; Müller Schmied et al., 2014; Müller Schmied et al., 2016). In its standard version, WaterGAP
does not take into account glaciers. To avoid the double-counting of glacierized areas, we integrated 0.5° gridded monthly
time series of LGWSA simulated by GGM as an input to WaterGAP. This resulted in a non-standard version of WaterGAP
which implicitly includes glaciers, hereafter referred to as integrated WaterGAP. The model was run with two different
precipitation forcings and two different assumptions regarding irrigation water use, resulting in an ensemble of four
solutions. We regarded the spread of these four time series around the ensemble mean as an informal indication of
uncertainty. We validated the ensemble by comparing it to an ensemble of four GRACE spherical harmonics (SH) solutions.

In the following section, we briefly describe the models and data sets used in this study. The methods employed in this study
are described in Section 3. In Sections 4 and 5, we present the results of our model evaluation and of our assessment of
global TWSA over the period 1948 to 2016, respectively. The results are discussed in Section 6. Finally, we present our
conclusions in Section 7.

## 2. Models and data

### 2.1 Global hydrological model

#### 2.1.1    General structure

We used the latest version of the GHM WaterGAP, WaterGAP2.2d. WaterGAP simulates both human water use as well as
daily water flows and water storages (or anomalies) on a 0.5° by 0.5° grid (55 km by 55 km at equator and ~3000 km$^2$ grid
cell) covering the global continental area except for Antarctica (see Fig. 1 in Döll et al., 2014). We ignored model outputs
over the grid cells corresponding to Greenland because this study focuses on anomalies over continental area different from
the ice sheets. Streamflow is laterally routed through the stream network derived from the global drainage direction map
DDM30 (Döll and Lehner, 2002) until it reaches the ocean or an inland sink. The model is calibrated against observations of
mean annual streamflow at 1319 gauging stations (Müller Schmied et al., 2014).





The model requires daily climatological input data sets of precipitation (rainfall and snowfall), near-surface air temperature
and long- and short-wave downwards surface radiation. In the frame of this study, we used an homogenized climate forcing
resulting from the combination of WATCH Forcing Data based on ERA-40 reanalysis (WFD, Weedon et al., 2011) for the
period 1948–1978 and WFD methodology applied to ERA-Interim reanalysis (WFDEI, Weedon et al., 2014) for the period
1979–2016 (Müller Schmied et al., 2016). Hereafter, we refer to this homogenized climate forcing simply as WFDEI.

Monthly sums of precipitation are bias corrected by monthly precipitation data sets derived from raingage observations of
either GPCC v5/v6 (Global Precipitation Climatology Centre, Schneider et al., 2015) or CRU TS3.10/TS3.21 (Climate
Research Unit, Harris et al., 2014). Note that the GPCC and CRU products used to scale monthly precipitation sums within
WFDEI use the available number of gauging stations for each month. The variability in the number of precipitation
observations over time makes the resulting precipitation data sets less suitable for trend analysis (Müller Schmied et al.,

2016). We forced WaterGAP with both WFDEI with monthly precipitation sums based on GPCC (hereafter WFDEI-GPCC)
and based on CRU (hereafter WFDEI-CRU) in order to account for part of the uncertainty in model output due to
precipitation input data.

WaterGAP was developed with the aim of assessing global water availability. Thus, one of its key features is the
representation of human impacts on water resources. Concretely, the model accounts for the impact of water impoundment

in reservoirs and of human water use on water flows and storages. $LWSA_{hum}$ is then calculated following Eq. (1):

$$LWSA_{hum} = LWSA_{res} + LWSA_{abs} \ , \hspace{5cm} (1)$$

where $LWSA_{res}$ is the anomaly due to impoundment of water in reservoirs and $LWSA_{abs}$ is the anomaly due to water
abstraction. Reservoir data (location, type, surface area, first operational year and storage capacity) used by the model comes
from a preliminary version of the Global Reservoir and Dam (GRanD) data base which includes 6862 reservoirs with a total

storage capacity of 6197 km³ (Lehner et al., 2011). The simulation of reservoir operation is based on the generic algorithm of
Hanasaki et al. (2006), which distinguishes between irrigation and non-irrigation reservoirs. A slightly modified version of
the algorithm (Döll et al., 2009) is implemented in WaterGAP. The model distinguishes between man-made reservoirs and
regulated lakes (i.e. natural lakes whose outflows are regulated by a dam). Reservoirs (man-made reservois plus regulated
lakes) are classified as "local" or "global" (the same distinction exists for wetlands and lakes). Local reservoirs are located

only within one grid cell and thus are fed only by runoff produced within that cell. Different from local reservoirs, global
reservoirs are spread within more than one cell and are also fed by streamflow from the upstream cell. In addition, global
reservoirs are defined by their size, exceeding a surface area of 100 km² or having a maximum storage capacity of at least 0.5
km³. Local lakes and local reservoirs within one cell are lumped into one local lake. This can be explained by the fact that
lumping multiple local reservoirs within one cell into one local reservoir inevitably erases the specific characteristics of each

reservoir; the resulting lumped local reservoir is then not expected to be better simulated by the reservoir algorithm then by
the lake algorithm (Döll et al., 2009). Taking this into account, 1082 global reservoirs and 85 global regulated lakes, which
together represent a total storage capacity of 5764 km³ (~16 mm SLE), are simulated by WaterGAP using the reservoir





operation algorithm. The reservoir filling phase upon construction is simulated based on the first operational year and the storage capacity. The monthly release flow of irrigation reservoirs varies according to the downstream consumptive water
use (i.e. part of water abstractions that evapotranspires during use). For non-irrigation reservoirs, it is assumed that the monthly release flow remains unchanged throughout the year.

Concerning human water use, in a first instance, time series of water abstraction and consumptive water use are generated for five water use sectors (irrigation, livestock farming, domestic use, manufacturing industries and cooling of thermal power plants) by separate global water use models. The calculation of irrigation water use takes into account climate variability as
well as yearly country estimates of irrigated area (Döll et al., 2012). The outputs of the water use models are then translated into net abstraction (i.e. total abstraction minus return flow) by the sub-model GWSWUSE, which distinguishes the source of abstracted water (surface water or groundwater). The net abstraction time series are then subtracted from the surface water and groundwater storage compartments of WaterGAP, respectively (Müller Schmied et al., 2014; Döll et al., 2014).

### 2.1.2 Computation of LWSA

WaterGAP represents the transport of water on continents as flows among a series of individual water storage compartments (see Figure 1 in Döll et al., 2014). It includes all continental water storage compartments except for glaciers. Thus, glacierized areas are treated as non-glacierized areas. LWSA, which integrates the changes in all the compartments, is calculated following Eq. (2):

$$LWSA = SnWSA + CnWSA + SMWSA + GWSA + LaWSA + ReWSA + WeWSA + RiWSA \, , \qquad (2)$$

where WSA is water storage anomaly in snow (Sn), canopy (Cn), soil moisture (SM), groundwater (G), lake (La), reservoir (Re), wetland (We) and river (Ri) storages.

### 2.2 Global glacier model

### 2.2.1 General structure

We used the global glacier model GGM of Marzeion et al. (2012). The model computes mass changes of individual glaciers
for the whole globe. It combines a glacier SMB model, following an empirically based temperature–index approach, with a model that accounts for the response of glacier geometry (in the model defined by area, length and elevation range) to changes in glacier mass. The dynamic simulation of this response follows an area–volume–time scaling approach, based on the equation of Bahr et al. (1997), enabling the model to account for various feedbacks between glacier geometry and mass balance. The model is calibrated by fitting simulated glacier SMB to observed glacier SMB from the collections of the
World Glacier Monitoring Service (2016). The error in modelled annual glacier mass change is determined using a cross-validation routine applied to glaciers with observed mass balances.

GGM is forced by global time series of near-surface air temperature and precipitation fluxes from various sources (Section 2.2.2). As initial conditions, it also requires information on glacier area and minimum and maximum elevation, which are





taken from the Randolph Glacier Inventory (RGI) version 6.0 (updated from Pfeffer et al., 2014). GGM includes both local

(i.e. glacier-specific) and global parameters. Local parameters are calibrated and cross-validated following the procedure described in Marzeion et al. (2012). Global parameters are optimized following a multi-objective optimization routine, maximizing temporal correlation of model results and observations, and minimizing the model bias as well as the difference of the variance of modelled and observed mass balances.

### 2.2.2  Computation of LGWSA

GGM computes glacier mass change at the scale of individual glaciers worldwide. In the framework of this study, the data were provided on a rectangular 0.5° by 0.5° grid (for consistency with WaterGAP spatial resolution) covering the entire globe (excluding the peripheral glaciers of Greenland and Antarctica) at monthly scale for the period extending from January 1948 to December 2016. Gridded annual time series of glacier area computed with GGM, as well as monthly time series of total (liquid plus solid) precipitation on glacier area from the atmospheric forcing were also used in this study. Glacier area

data was required to adapt the land area fraction within WaterGAP cells. In addition, precipitation on glacier area was required to calculate glacier runoff (see Section 3.3 for more details). Note that, to produce the gridded GGM data sets, each glacier was assigned to the grid cell that contains its center point (as given in the RGI version 6.0) even if, in reality, the glacier stretches across several grid cells. The data sets used in this study correspond to a simulation in which GGM was forced by the mean of an ensemble of seven global gridded atmospheric data sets (New et al., 2000; Saha et al., 2010;

Compo et al., 2011; Dee et al., 2011; Kobayashi et al., 2015; Poli et al., 2016; Gelaro et al., 2017). Note that choosing the ensemble mean over any of the individual atmospheric data sets allows reducing the uncertainty due to input climate forcing data.

### 2.3  GRACE-derived TWSA

Global time series of GRACE mass change over the continents were derived from ITSG-Grace2018 (Mayer-Gürr et al.,

2018) and GRACE Release 6 (CSR, GFZ, JPL) quasi-monthly Level-2 gravity field solutions by means of global spherical harmonic (SH) coefficients. We further processed the SH solutions in order to derive global grids of surface mass change, as described below.

SH unconstrained monthly solutions expanded up to degree and order 60 were chosen for the lower noise level (compared to higher resolved solutions). We substituted Degree-1 (geocenter motion) coefficients following the approaches of Swenson et

al. (2008) and Bergmann-Wolf et al. (2014), and $C_{2,0}$ (Earth's oblateness) coefficients after Cheng et al. (2013) for the coefficients in the GRACE products, respectively. Mass redistribution related to glacial isostatic adjustment (GIA) was accounted for via the GIA modelling results from Caron et al. (2018). Furthermore, we excluded areas with considerable mass redistribution related to the 2004/2005 Sumatra/Nias- and the 2011 Tōhoku earthquakes from the integration. We corrected for the leakage effect from continent to ocean, observed in coastal regions, by expanding the initial land–water

mask by a 300 km buffer onto the ocean. The gravity field over this buffer area contains signal from both land and ocean. In





order to counteract this superposition, we subtracted the monthly mean value of the buffered Global Ocean surface-density change (obtained from the corresponding SH solution) multiplied by the fractional ocean area of the buffer cells, respectively. Here, we assume the actual mean OMC over the buffer to equal the global mean OMC.

The resulting integrated and corrected signal was then attributed to the initial land–water mask (i.e. the one used by
WaterGAP) area and represents the global continental mass change from hydrology and glaciers (since it is impossible for GRACE to make the distinction), excluding Antarctica and Greenland. Note that mass changes are given relative to an arbitrary reference state; here we use the temporal mean state over the period January 2006 to December 2015. All four solutions (i.e. based on ITSG-Grace2018 and CSR, GFZ and JPL Release 6 products) cover the period from January 2003 to August 2016, with some months with missing data in between.

The GRACE trend uncertainty is a 1σ standard uncertainty and was assessed from several components in the time series processing that have a significant impact on the trend. It comprises uncertainty due to leakage corrections, low-degree replacements for GRACE (Earth flattening and geocenter motion) and GIA. The given combined trend uncertainty, which is identical for all GRACE time series, is the root sum squared of these contributing parts.

### 3. Methods

### 3.1 Evaluation of GGM glacier mass change with in situ observations

Marzeion et al. (2012) compared in situ observations of glacier annual SMB (from the collections of the World Glacier Monitoring Service) to estimates obtained with GGM. The results showed a good correlation (r = 0.60 for set of 3997 pairs of annual modelled and measured SMB) and from this it is concluded that GGM can reconstruct annual mass balances of unmeasured glaciers with reasonable accuracy. In this study, which focuses not only on the annual but also on the monthly
scale, we investigated the performance of GGM at seasonal scale. By "seasonal", we refer to the winter and summer seasons within a glacier mass balance year. During the winter (accumulation) season (October to March in the Northern hemisphere and April to September in the Southern hemisphere), the glacier tends to gain mass, while during the summer (melting) season (April to September in the Northern hemisphere and October to March in the Southern hemisphere), the glacier tends to lose mass. For the comparison, we used the estimates computed by GGM at the scale of individual glaciers. The
observational data was selected from the "reference glaciers" sample of the World Glacier Monitoring Service (2017). This constitutes a reliable and well-documented sample of globally distributed long-term observation series. A glacier was selected from this sample if 1) its observations corresponded to the entire glacier and not solely to sections of it, 2) it had a minimum of five years with observations for both summer and winter and 3) it was among the glaciers simulated by GGM. In total, 31 glaciers worldwide were selected (see Table S1 in the supplementary material).

### 3.2 Pre-processing of gridded GGM output data

We applied a number of pre-processing steps to the GGM gridded data in order to make it suitable as input data for WaterGAP. For all three data sets (temporally accumulated glacier mass change, glacier area and precipitation on glacier





area), the grid cells containing one or more glaciers (hereafter "glacierized" cells) were assigned to the corresponding cells of the WaterGAP grid based on geographical location. As a consequence, 49 GGM glacierized cells were excluded from the

analysis as they were outside the WaterGAP grid boundaries (see Figure S1 in the supplementary material). Note that the accumulated global glacier mass change over 1948–2016 from the original data set (i.e. including all glacierized cells) amounts to 5730.63 Gt; after the exclusion of the 49 cells, it amounts to 5715.81 Gt. Hence, we consider that the effect of excluding these cells is negligible. Furthermore, the accumulated glacier mass change time series were disaggregated from monthly to daily because WaterGAP runs internally at daily time steps. At the scale of one month, the temporal

disaggregation was done by assigning the accumulated monthly value to the last day of the month and subsequently replacing the missing values (rest of the days in the month) by simple linear interpolation. Note that the first day of the simulation period was assigned the value 0. The next step consisted in deriving the rate of daily glacier mass change from the accumulated daily values. Time series of precipitation on glacier area were also disaggregated from monthly to daily; at the scale of one month, this was done by simply dividing the monthly precipitation by the number of days in the month and

assigning the resulting precipitation to each day of the month.

### 3.3  Integration of GGM glacier data into WaterGAP

Each WaterGAP grid cell has a continental area ($A_C$), the part of the grid cell that is not ocean. The continental area consists of spatially and temporally varying fractions of land area ($A_L$) (where precipitation infiltrates into the soil) and areas of SWB if there are lakes, wetlands and/or reservoirs. If a fraction of $A_L$ is actually covered by glacier according to GGM, then the

simulation of hydrological processes (evapotranspiration, runoff generation, groundwater recharge etc.) needs to be restricted, in the integrated WaterGAP version, to the glacier-free portion of the land area. In the gridded glacier area ($A_{LG}$) data set, the entire area of each glacier is assigned to the cell where the center of the glacier is located. However, in reality, some glaciers are spread over more than one cell; this means that sometimes input $A_{LG}$ is larger than $A_C$. In such cases, $A_{LG}$ is set to be equal to $A_C$, in order to avoid inconsistencies. As a result of this adaptation, we systematically neglect 10 to 11%

(depending on the year) of the global $A_{LG}$ over the period 1948–2016 (but not the pertaining LGWSA). The adapted $A_{LG}$ is then used to adjust $A_L$ (Figure 1). In the initial year of the simulation (yr = 1), $A_L$ (which is equal to the initial $A_L$ of the standard WaterGAP) is reduced by $A_{LG}$. In the following years, $A_L$ is adapted by the glacier area change ($\Delta A_{LG}$), which can be either positive (area increase) or negative (area decrease). Areas of SWB are not adapted according to $A_{LG}$. Glacier mass change (dLGWS/dt) computed by GGM is added, along with changes in the other storage compartments, to TWS change

(Figure 1). We assume that the only ongoing hydrological process on $A_{LG}$ is runoff generation from precipitation on the glacier ($P_{LG}$) and the LGWS change (dLGWS/dt), hereafter called glacier runoff ($R_{LG}$) and calculated according to Eq. (3):

$$R_{LG} = P_{LG} - dLGWS/dt \qquad\qquad (3)$$

If daily increase in glacier mass is larger than daily $P_{LG}$, $R_{LG}$ is assumed to be equal to zero. $R_{LG}$ is added to the cell's fast runoff, which partly flows directly into the river storage and partly into the lake, wetland and reservoir storages (Figure 1). It





is assumed that $R_{LG}$ does not recharge the soil and groundwater storages. The thus enhanced WaterGAP, the "integrated"
WaterGAP, is capable of actually simulating TWSA on the continents (as observable by GRACE), while the standard
WaterGAP neglects the impact of glaciers on TWSA.

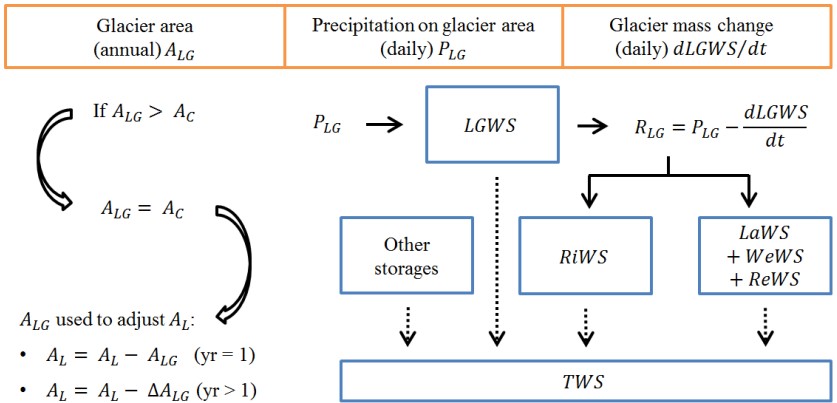

**Figure 1:** Schematic of integration of glacier data from GGM into WaterGAP at grid cell scale. Glacier-related data sets are represented
by orange boxes. Blue boxes represent water storage compartments in WaterGAP. Full arrows represent water flows and dotted arrows
indicate that the sum of changes in individual storages equals TWS change. See text for details and abbreviations.

### 3.4 Overview of WaterGAP variants

In order to generate the monthly and annual TWSA time series used in this study, two model versions, standard WaterGAP
(Wg_std) and integrated WaterGAP (Wg_gl), were run under four different model configurations or modes (Table 1). In
anthropogenic mode (standard mode), the model takes into account both climate variability and anthropogenic variability,
the latter referring to the effects of water impoundment in reservoirs and of human water abstraction. In naturalized mode,
the model takes into account only climate variability; this means that reservoirs are not simulated at all (except for regulated
lakes, which are treated as natural lakes) and water abstraction is neglected. By comparing outputs from anthropogenic and
naturalized runs, it is possible to isolate the water storage change solely due to the human activities (Table 2). WaterGAP
also allows performing runs that neglect reservoirs but take into account water abstraction, and vice-versa; these model
variants are interesting for the purpose of isolating the effect of water impoundment in reservoirs from the effect of water
abstraction (Table 2). Each combination of model version and model configuration was run under two climate forcings,
WFDEI-GPCC and WFDEI-CRU.






**Table 1:** Overview of model variants used in this study. The standard version of WaterGAP2.2d (Wg_std) and a non-standard version which implicitly includes glaciers (Wg_gl) were run under four types of model configuration ("anthropogenic", "anthropogenic without reservoirs", "anthropogenic without water abstraction" and "naturalized"), two climate forcings differing in terms of precipitation bias correction (based on GPCC or CRU), and two assumptions related to consumptive irrigation water use ("70% deficit" and "optimal").

| Model version | Model configuration | Precipitation bias correction | Consumptive irrigation water use | Model variant name |
|---|---|---|---|---|
| Standard WaterGAP (Wg_std) | Anthropogenic | GPCC[1] | 70% deficit | Wg_std_ant_GPCC_irr70 |
| | | | optimal | Wg_std_ant_GPCC_irr100 |
| | Anthropogenic | CRU[2] | 70% deficit | Wg_std_ant_CRU_irr70 |
| | | | optimal | Wg_std_ant_CRU_irr100 |
| Integrated WaterGAP (Wg_gl) | Anthropogenic | GPCC[1] | 70% deficit | Wg_gl_ant_GPCC_irr70 |
| | | | optimal | Wg_gl_ant_GPCC_irr100 |
| | Anthropogenic without reservoirs | GPCC[1] | 70% deficit | Wg_gl_ant_nores_GPCC_irr70 |
| | | | optimal | Wg_gl_ant_nores_GPCC_irr100 |
| | Anthropogenic without water abstraction | GPCC[1] | | Wg_gl_ant_noabs_GPCC |
| | Naturalized | GPCC[1] | | Wg_gl_nat_GPCC |
| | Anthropogenic | CRU[2] | 70% deficit | Wg_gl_ant_CRU_irr70 |
| | | | optimal | Wg_gl_ant_CRU_irr100 |
| | Anthropogenic without reservoirs | CRU[2] | 70% deficit | Wg_gl_ant_nores_CRU_irr70 |
| | | | optimal | Wg_gl_ant_nores_CRU_irr100 |
| | Anthropogenic without water abstraction | CRU[2] | | Wg_gl_ant_noabs_CRU |
| | Naturalized | CRU[2] | | Wg_gl_nat_CRU |

[1] Schneider et al. (2015). [2] Harris et al. (2014).

In addition, inspired by the study of Döll et al. (2014), we considered two different assumptions with respect to consumptive irrigation water use in the design of the variants. Consumptive irrigation water use is normally computed under the assumption that crops receive enough irrigation water to allow actual evapotranspiration to become equal to the potential evapotranspiration value (Döll et al., 2016). However, in reality, this is not always the case, especially in regions affected by groundwater depletion (GWD), where farmers may use less water due to water scarcity. GWD is defined as a long-term decline of hydraulic heads and groundwater storage. Using a former version of WaterGAP, Döll et al. (2014) identified GWD areas worldwide by selecting the grid cells characterized by 1) an average GWD of at least 5 mm yr$^{-1}$ over the period 1980–2009 and 2) an irrigation water abstraction volume of at least 5% of total water abstraction volume. In this study, we applied two different assumptions in relation to consumptive irrigation water use in the GWD areas identified by Döll et al.



(2014). We either assumed that consumptive irrigation water use is optimal (i.e. that it corresponds to 100% of water requirement) or that it is equal to 70% of optimal (Table 1). Hereafter, we refer to these two assumptions as "optimal irrigation" and "70% deficit irrigation", respectively.

**Table 2:** Overview of how the TWSA mass budget components were calculated using the integrated WaterGAP variants. TWSA[ant], TWSA[nat], TWSA[ant_nores] and TWSA[ant_noabs] were computed under the 'anthropogenic', 'naturalized', 'anthropogenic without reservoirs' and 'anthropogenic without water abstraction' configurations, respectively. Land glacier water storage anomalies (LGWSA) remain unchanged throughout the configurations.

| Component | Computation | Model configuration(s) used |
|---|---|---|
| LWSA | TWSA[ant] – LGWSA | 'Anthropogenic' |
| $LWSA_{clim}$ | TWSA[nat] – LGWSA | 'Naturalized' |
| $LWSA_{hum}$ | TWSA[ant] – TWSA[nat] | 'Anthropogenic' and 'naturalized' |
| $LWSA_{res}$ | TWSA[ant] – TWSA[ant_nores] | 'Anthropogenic' and 'anthropogenic without reservoirs' |
| $LWSA_{abs}$ | TWSA[ant] – TWSA[ant_noabs] | 'Anthropogenic' and 'anthropogenic without water abstraction' |

## 4. Model evaluation

To evaluate the quality of GGM, simulated glacier mass changes of individual glaciers were compared to glacier observations. Then, global mean TWSA simulated by WaterGAP with and without integration of GGM output was compared to GRACE observations.

### 4.1 Comparison of observed and simulated annual and seasonal glacier mass changes

Comparison of observed average annual glacier mass changes for 31 glaciers with mostly decades of observations (Table S1) confirm the conclusion of Marzeion et al. (2012) that GGM is able to simulate well annual glacier mass changes. This study reveals that both winter accumulation and summer melting is simulated reasonably, too, but worse than the annual mass changes, with Nash–Sutcliffe efficiencies NSE (Eq. (1), Nash and Sutcliffe, 1970) and correlation coefficients r being slightly lower than for the annual values (Figure 2). We also quantified, for each glacier, the fit between simulated and

observed time series of winter and summer mass changes (two values per year times the number of years with observations). Approximately three-quarters of the glaciers have a NSE higher than 0.70 (Table S1), indicating a good model performance at the seasonal time scale even though GGM was only tuned with respect to the annual values. Only two glaciers, the "Devon Ice Cap NW" and the "Vernagtferner", show a negative NSE. The first is a marine-terminating ice cap where calving processes that are not modelled explicitly by GGM occur.





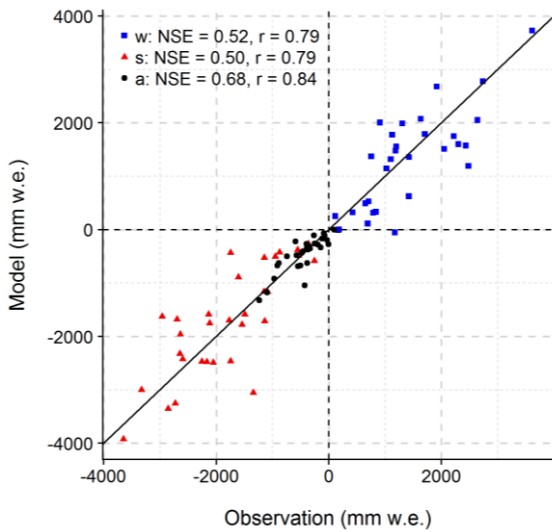


**Figure 2:** Correlation between observed and modelled average annual, winter and summer glacier mass change. Observations were taken from the collections of the World Glacier Monitoring Service (31 glaciers included). Model results were obtained with the global glacier model of Marzeion et al. (2012). Nash–Sutcliffe efficiency (NSE) and correlation coefficient (r) values correspond to average annual (a), winter (w) and summer (s) mass changes. Millimetres of water equivalent (mm w.e.) are relative to glacier area.

**4.2 Comparison of observed and simulated global mean TWSA during the period January 2003 to August 2016**

Figure 3a presents time series of the ensemble of monthly TWSA simulated by the standard (without glaciers) and the integrated (with glaciers) WaterGAP versions compared to GRACE observations. The NSE- and r-values shown in the figure were computed for the mean of the GRACE ensemble (consisting of four solutions) and the means of the Wg_std and Wg_gl ensembles (each ensemble consisting of the four variants under anthropogenic conditions, see Table 1). For both

Wg_std and Wg_gl, there is a remarkably good fit between the modelled ensemble mean and the GRACE ensemble mean in terms of NSE (0.85, 0.87) and r (0.92, 0.95), both of which rather reflect the good fit of seasonal variability than of the trend. The fit is slightly better with Wg_gl, not only in terms of ensemble mean but also if we consider the NSE- and r-values obtained by comparing each individual GRACE solution to each individual WaterGAP solution (see Figure S2 in the supplementary material). With NSE around 0.80 during the period January 2003 to December 2008, the fit is worse during

the first six evaluation years than during the following period until August 2016 (Figure 3a). Note however that the period from 2011 onward contains more gaps in the GRACE data. Glaciers lead to a much stronger decreasing TWSA trend over the period considered. Monthly time series of LGWSA from GGM have a small seasonal variability and an almost linear decreasing trend (Figure 3b). When adding LGWSA to the LWSA computed by Wg_std (Wg_std+GGM), the resulting time series of global mean values (purple ensemble in Fig. 3b) is indistinguishable from the TWSA time series computed by

Wg_gl (green ensemble in Fig. 3a).





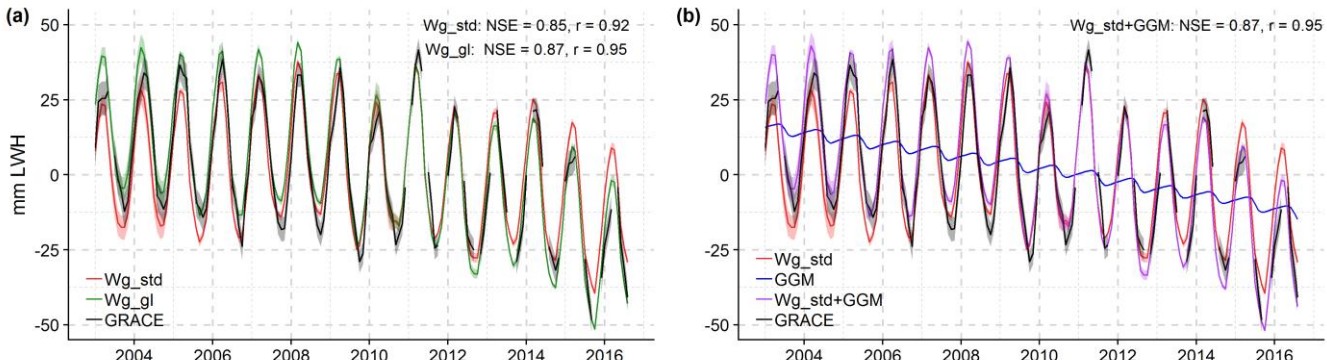

**Figure 3:** Global mean monthly TWSA from GRACE observations and from different modelling approaches, January 2003 to August 2016. (a) TWSA from GRACE ensemble, LWSA from standard WaterGAP (Wg_std) in anthropogenic mode ensemble (Wg_std_ant_CRU_irr100, Wg_std_ant_CRU_irr70, Wg_std_ant_GPCC_irr100 and Wg_std_ant_GPCC_irr70 in Table 1) and TWSA from integrated WaterGAP (Wg_gl) in anthropogenic mode ensemble (Wg_gl_ant_CRU_irr100, Wg_gl_ant_CRU_irr70, Wg_gl_ant_GPCC_irr100 and Wg_gl_ant_GPCC_irr70 in Table 1). (b) TWSA from GRACE ensemble, LWSA from Wg_std ensemble as in (a), LGWSA from GGM and TWSA obtained by adding anomalies from Wg_std ensemble and GGM (Wg_std+GGM). For each ensemble, the curve represents the ensemble mean and the shaded area around the curve represents either the uncertainty range (GRACE) or the ensemble minimum and maximum values (Wg_std, Wg_gl and Wg_std+GGM). Nash–Sutcliffe efficiency (NSE) and correlation coefficient (r) obtained by comparing GRACE and model ensemble means are provided. Anomalies are relative to the mean over the period January 2006 to December 2015. Millimetres of land water height (mm LWH) are relative to the global continental area without the ice sheets ($132.3 \cdot 10^6$ km$^2$).

To evaluate WaterGAP performance separately regarding its simulation of seasonality, trend and inter-annual variability, the original monthly TWSA time series (Figure 3a) were decomposed (based on harmonic analysis) into de-trended (Figures 4a and 4b), de-seasonalized (Figures 4c and 4d) and residual (Figures 4e and 4f) TWSA. Regarding seasonality, there is a remarkably good fit to GRACE with both Wg_std and Wg_gl; the seasonal amplitude and phase are very well reproduced by the models, even though for some years (e.g. 2003, 2004, 2011, 2014) there seems to be a slight phase shift of approximately one month (Figures 4a and 4b). The indicators show that the fit is very good with the two models and only slightly better for Wg_gl, reflecting the small contribution of glaciers to the seasonal variation of global mean TWSA (NSE of 0.89 instead of 0.88 due to slightly increasing seasonal amplitude). The de-seasonalized time series show the strong impact of including glaciers into WaterGAP. Wg_gl can follow the decrease of TWSA observed by GRACE much better than Wg_std (Figures 4c and 4d), and performance indicators are significantly higher (NSE improves from 0.65 to 0.74 and r from 0.85 to 0.93).

**Figure 4:** Temporal components of global mean monthly TWSA from GRACE observations and from two versions of WaterGAP2.2d,
January 2003 to August 2016. GRACE ensemble, standard WaterGAP (Wg_std) ensemble (a, c, e) and integrated WaterGAP (Wg_gl)
ensemble (b, d, f). (a,b) De-trended anomalies. (c,d) De-seasonalized anomalies (correspond to linear and non-linear long-term variability).
(e,f) Residual anomalies obtained by removing linear trend and seasonality (correspond to non-linear inter-annual variability). For each
ensemble, the curve represents the ensemble mean and the shaded area around the curve represents either the uncertainty range (GRACE)
or the ensemble minimum and maximum values (Wg_std and Wg_gl). Nash–Sutcliffe efficiency (NSE) and correlation coefficient (r)
obtained by comparing GRACE and model ensemble means are provided. Anomalies are relative to the mean over the period January
2006 to December 2015 and given in millimetres of land water height (mm LWH).





However, the GRACE signal is overestimated before 2011(in particular from 2007–2009) and in 2016, and underestimated in 2011. The overestimation from 2007–2009 may be partly due to a drought period in the Near East when a large number of new groundwater wells were drilled in this region, which is not taken into account in WaterGAP simulations of groundwater

vs. surface water use (Döll et al., 2014). The residual signal present in the original time series (Figures 4e and 4f), which includes the inter-annual variability, is very similar for the two models, which suggests that GGM does not contribute to the residual. The fit of the residuals and thus simulation of inter-annual variability is relatively good but worse than for de-trended and de-seasonalized time series. The discrepancies to the GRACE signal follow the same pattern as in Figures 4c and 4d. However, the fit to GRACE before 2007 is better than in the latter.

Linear trends are very sensitive to the selected time period and individual values. While the de-seasonalized TWSA from Wg_gl fits reasonably well overall to GRACE observations (Fig. 4d), Wg_gl considerably overestimates the trend determined for the time period January 2003 to August 2016, if averaged over the four ensemble members, by about 30% (Table 3). Wg_std variants underestimate the positive contribution to OMC by about 50%. Thus, the TWSA trend computed by integrating GGM output into WaterGAP results in a better estimation of the GRACE trend than if glaciers are neglected.

Assuming 70% deficit irrigation and utilizing GPCC precipitation, the simulated trend value of 1.05 mm SLE yr$^{-1}$ is within the uncertainty bounds of the GRACE solutions (Table 3). The trend gets larger with optimal irrigation and CRU precipitation, which is mainly due to the larger TWSA values during the period 2003–2004 (Figure 4d). The absolute difference between the two irrigation variants (0.11 to 0.12 mm SLE yr$^{-1}$) is in accordance with the absolute difference between the two precipitation forcings (0.11 to 0.13 mm SLE yr$^{-1}$); this means that, over this period, the trend is equally

affected by the choice of irrigation variant than by the choice of precipitation forcing. The GRACE ensemble range is approximately 5 times smaller than the range of the Wg_std and Wg_gl ensembles. This is partly due to the choice of the GRACE solutions; although coming from different processing centers, they were all corrected using the same GIA model (Caron et al., 2018). The trend-spread owing to possible GIA models is reflected in the given standard uncertainty. The GIA model choice is the main contributor to uncertainty besides the GRACE degree-1 correction.

Overall, we infer that integration of glacier model output into WaterGAP results in a better fit to GRACE in terms of linear trend, NSE and r for de-seasonalized time series while simulated seasonality is barely improved. Seasonal variations of mean global TWSA are simulated very well with Wg_gl compared to GRACE observations, inter-annual variability is captured to a certain extent, and the linear trend is overestimated. The overall fit of the monthly time series of global mean TWSA to GRACE (Figure 3a) is remarkably good (NSE = 0.87, r = 0.95). Together with the positive evaluation of the glacier model

results, this gives us confidence that our modelling approach can be used to reconstruct TWSA and thus the water mass transfer from the continents to the oceans for the time period before GRACE.



**Table 3:** Linear TWSA trends from GRACE observations and from WaterGAP2.2d, January 2003 to August 2016. Model estimates
correspond to individual solutions of standard WaterGAP (Wg_std) and integrated WaterGAP (Wg_gl) ensembles. GRACE-derived
estimates correspond to SH solutions from four processing centres (CSR, GFZ, ITSG and JPL). Trends were calculated according to the
linear least squares regression method. Negative trends (mass loss) over the continents, expressed in millimetres of land water height (mm
LWH, relative to the global continental area without the ice sheets $132.3 \cdot 10^6$ km$^2$), translate to positive trends (mass gain) over the oceans,
expressed in millimetres of sea level equivalent (mm SLE, relative to the global ocean area $361.0 \cdot 10^6$ km$^2$).

| | Variant | Trend | | Average of individual trends | |
|---|---|---|---|---|---|
| | | mm LWH yr$^{-1}$ | mm SLE yr$^{-1}$ | mm LWH yr$^{-1}$ | mm SLE yr$^{-1}$ |
| Wg_std | ant_GPCC_irr70 | -0.78 | 0.29 | -1.12 | 0.41 |
| | ant_GPCC_irr100 | -1.12 | 0.41 | | |
| | ant_CRU_irr70 | -1.13 | 0.42 | | |
| | ant_CRU_irr100 | -1.45 | 0.53 | | |
| Wg_gl | ant_GPCC_irr70 | -2.86 | 1.05 | -3.18 | 1.17 |
| | ant_GPCC_irr100 | -3.19 | 1.17 | | |
| | ant_CRU_irr70 | -3.18 | 1.16 | | |
| | ant_CRU_irr100 | -3.50 | 1.28 | | |
| GRACE | CSR_rl06sh | -2.37 ± 0.55 | 0.87 ± 0.20 | -2.37 ± 0.55 | 0.87 ± 0.20 |
| | GFZ_rl06sh | -2.39 ± 0.55 | 0.87 ± 0.20 | | |
| | ITSG_2018 | -2.29 ± 0.55 | 0.84 ± 0.20 | | |
| | JPL_rl06sh | -2.43 ± 0.55 | 0.89 ± 0.20 | | |

## 5. Global water transfer from continents to oceans over the period 1948–2016

Annual time series of global mean water storage anomalies over the period 1948–2016 were calculated by the four integrated
WaterGAP in anthropogenic mode variants with different precipitation inputs and irrigation water assumptions (Figure 5a).
The continents lost between 93–111 mm LWH to the oceans between 1948 and 2016, which is equivalent to an ocean water
mass increase of 34–41 mm SLE. SLR is less pronounced in case of less than optimal irrigation in groundwater depletion
areas (variant _irr70, see also Table 4). While the 2003–2016 trends are equally affected by the precipitation data set and the
irrigation assumption (Table 3), the 1948–2016 trends are much more strongly affected by the irrigation assumption than the
applied precipitation data set (Table 4). Continental water mass losses have been accelerating over time (see Table 5 and
Table S2 in the supplementary material) so that the 2003–2016 trends are approximately 6 times larger than the 1948–1975
trends.

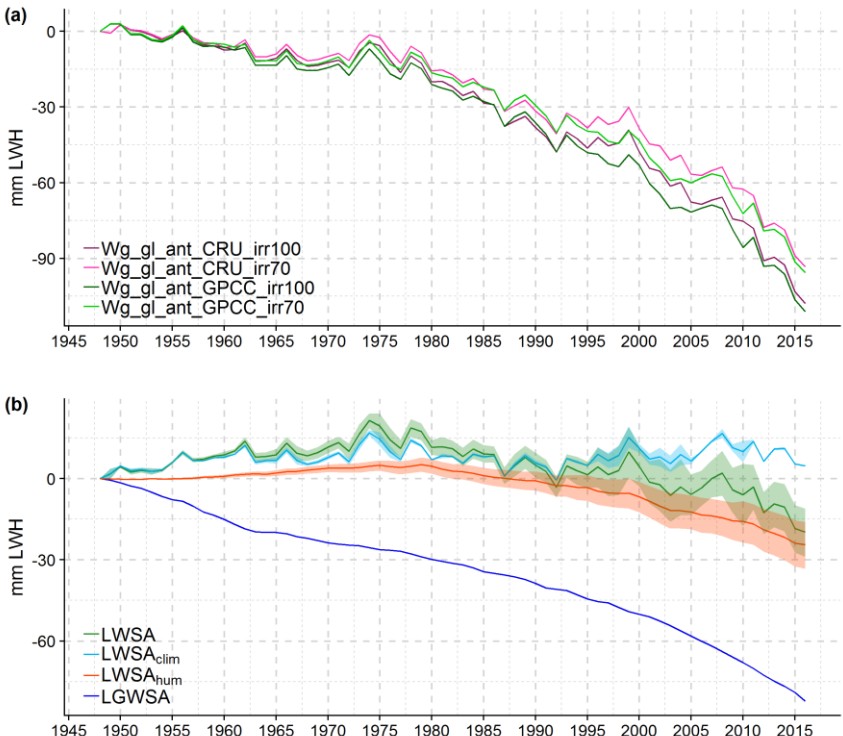

**Figure 5:** Global annual TWSA and individual contributions, 1948 to 2016. (a) TWSA computed with four variants of integrated WaterGAP in anthropogenic mode (Table 1). (b) Disaggregation of TWSA into anomalies of land glacier water storage (LGWSA) and land water storage (LWSA), and of land water storage anomalies into anomalies of climate-driven land water storage (LWSA$_{clim}$) and human-driven land water storage (LWSA$_{hum}$); for each ensemble, the curve represents the ensemble mean and the shaded area around the curve represents the ensemble minimum and maximum values. Anomalies are relative to the year 1948 and given in millimetres of land water height (mm LWH).

**Table 4:** Linear trends of contribution of TWSA to OMC, 1948 to 2016. Estimates were obtained with four variants of integrated WaterGAP (same variants as in Figure 5a). Positive trends translate to ocean mass gain, whereas negative trends translate to ocean mass loss. Estimates are given in millimetres of sea level equivalent per year (mm SLE yr$^{-1}$).

| Model variant | Linear trend mm SLE yr$^{-1}$ |
|---|---|
| Wg_gl_ant_GPCC_irr70 | 0.44 |
| Wg_gl_ant_GPCC_irr100 | 0.53 |
| Wg_gl_ant_CRU_irr70 | 0.42 |
| Wg_gl_ant_CRU_irr100 | 0.50 |





### 5.1 Contributions of LGWSA, climate-driven LWSA and human-driven LWSA to TWSA

Simulated TWSA (Figure 5a) was disaggregated into its individual components LGWSA, LWSA$_{clim}$ and LWSA$_{hum}$ (Figure 5b) by using the results of the different Wg_gl model variants (Tables 1 and 2). Glacier mass loss is the dominant component of the TWSA mass budget (Figure 5b), with LGWSA accounting for 81% of the cumulated water mass loss from continents over 1948–2016. The overall contribution of LWSA, which is dominated by its human-driven component (Figure 5b), is also positive (19% of the cumulated water mass loss). Inter-annual variability of LWSA stems from its climate-driven component (Figure 5b). Trends of TWSA, LGWSA, LWSA, LWSA$_{clim}$ and LWSA$_{hum}$ show an acceleration of continental water mass loss over time (Table 5). However, note that LWSA$_{clim}$ and LWSA$_{hum}$ exhibit negative contributions to OMC over the period 1948–1975, adding some water to the continents due to climate and human activities.

### 5.2 Contribution of reservoirs and water abstraction to human-driven LWSA

LWSA$_{hum}$ can be disaggregated into changes due to reservoir construction and operation (LWSA$_{res}$) and changes due to human water abstraction (LWSA$_{abs}$). Between 1948 and 2016, the continents gained approximately 22 mm LWH (i.e. 8 mm SLE) due to water impoundment in reservoirs, and lost between 40–57 mm LWH (i.e. 15–21 mm SLE) due to water abstraction for human water use, resulting in an overall positive contribution of LWSA$_{hum}$ to OMC (Figure 6).

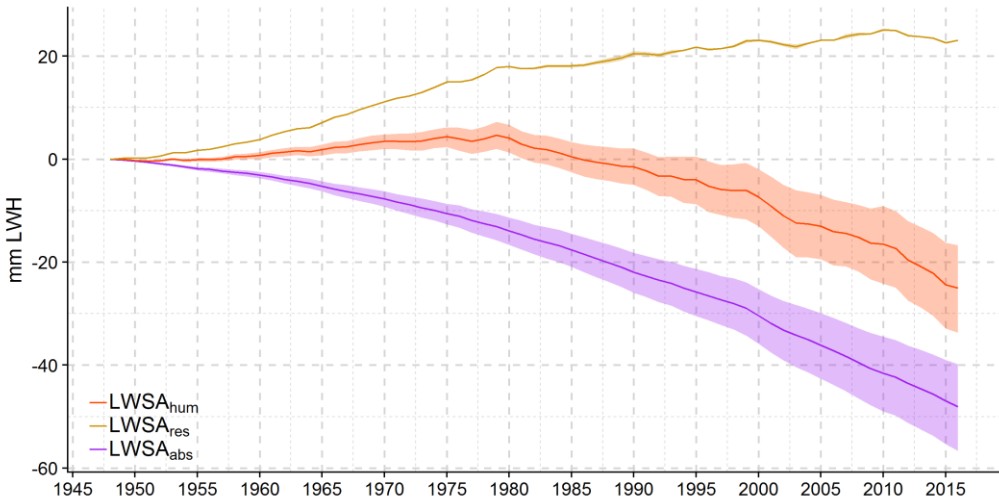

**Figure 6:** Global mean annual human-driven LWSA and individual contributions, 1948 to 2016. Human-driven LWSA (LWSA$_{hum}$, as in Figure 5) are disaggregated into anomalies due to reservoir operation (LWSA$_{res}$) and water abstraction (LWSA$_{abs}$). Anomalies are relative to the year 1948 and given in millimetres of land water height (mm LWH).

However, continental water mass gain due to LWSA$_{res}$ more than compensated mass losses due to LWSA$_{abs}$ before 1980, when intensive reservoir construction lead to a stronger increase of impounded water mass than afterwards (Figure 6). Trends of LWSA$_{res}$ show a deceleration of continental mass gain due to water impoundment in reservoirs over time, whereas trends of LWSA$_{abs}$ show an acceleration of continental mass loss due to water abstraction (Table 5).





**Table 5:** Linear trends of contribution of TWSA, LWSA, LWSA$_{clim}$, LWSA$_{hum}$, LWSA$_{res}$, LWSA$_{abs}$ and individual water storage compartments to OMC over three periods (1948–1975, 1976–2002 and 2003–2016). Positive trends translate to ocean mass gain, whereas negative trends translate to ocean mass loss. Ensemble ranges are given in parentheses. Estimates are given in millimetres of sea level equivalent per year (mm SLE yr$^{-1}$).

| Component | Linear trend mm SLE yr$^{-1}$ | | |
|---|---|---|---|
| | 1948–1975 | 1976–2002 | 2003–2016 |
| TWSA | 0.18 | 0.58 | 1.18 |
| | (0.13 to 0.23) | (0.49 to 0.66) | (1.06 to 1.30) |
| LWSA | -0.20 | 0.21 | 0.41 |
| | (-0.25 to -0.15) | (0.12 to 0.29) | (0.29 to 0.52) |
| LWSA$_{clim}$ | -0.13 | 0.01 | 0.04 |
| | (-0.15 to -0.10) | (-0.02 to 0.04) | (-0.03 to 0.10) |
| LWSA$_{hum}$ | -0.08 | 0.19 | 0.37 |
| | (-0.10 to -0.05) | (0.14 to 0.25) | (0.30 to 0.45) |
| LWSA$_{res}$ (= ReWSA) | -0.21 | -0.10 | -0.02 |
| | | (-0.11 to -0.10) | (-0.03 to -0.02) |
| LWSA$_{abs}$ | 0.14 | 0.30 | 0.39 |
| | (0.12 to 0.17) | (0.25 to 0.35) | (0.33 to 0.46) |
| LGWSA | 0.38 | 0.37 | 0.77 |
| CnWSA | 0.00 | 0.00 | 0.00 |
| SnWSA | -0.06 | 0.01 | 0.03 |
| SMWSA | -0.01 | 0.01 | -0.01 |
| | | (0.00 to 0.02) | (-0.03 to 0.00) |
| GWSA | 0.11 | 0.26 | 0.39 |
| | (0.08 to 0.14) | (0.21 to 0.32) | (0.32 to 0.46) |
| LaWSA | -0.02 | 0.00 | -0.01 |
| | (-0.02 to -0.01) | | (-0.01 to 0.00) |
| WeWSA | -0.01 | 0.01 | -0.01 |
| | (-0.01 to 0.00) | (0.00 to 0.01) | (-0.01 to 0.00) |
| RiWSA | 0.00 | 0.01 | 0.03 |
| | (-0.01 to 0.01) | (0.00 to 0.02) | (-0.01 to 0.07) |


### 5.3 Contribution of individual water storage compartments to TWSA

Among the nine water storage compartments in Wg_gl, largest absolute change over the period 1948–2016 is mass loss from glaciers, i.e. a decrease of LGWS by 30 mm SLE (Figure 7a). The second largest is groundwater depletion, with a decrease of 13–19 mm SLE depending on the irrigation assumption. The third largest (with opposite sign) is water impoundment in





reservoirs, which added 8 mm SLE to the continents. In the storages of SWB, there are only very small differences due to the irrigation variant. Apart from LGWS, GWS and ReWS, the rest of the water storage compartments have only a marginal contribution, with negative contributions from the river and wetland storages, and positive contributions from the soil, lake and snow storages (Figure 7b). Differences related to precipitation forcing, which are more visible in Figure 7b, exist in all storages except for the glacier storage, which is not affected by the different WaterGAP precipitation forcings as it is a direct

input to WaterGAP.

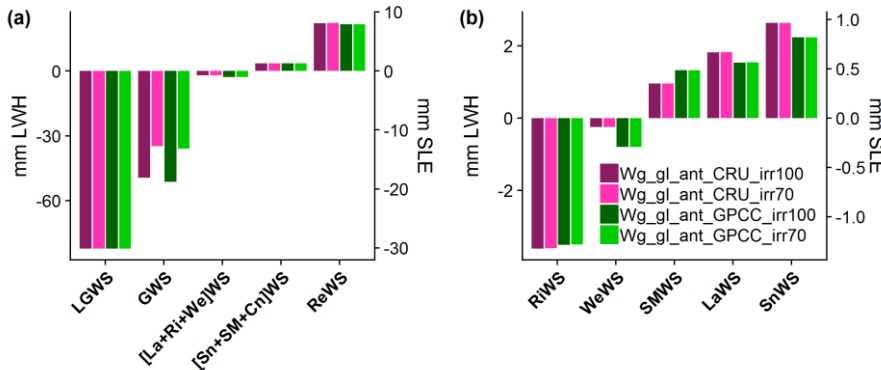

**Figure 7:** Global cumulated water storage change in individual water storage compartments, 1948 to 2016. Estimates were obtained with four variants of integrated WaterGAP in anthropogenic mode (Table 1). (a) Water storage (WS) change in glacier (LG), groundwater (G), aggregate of lake (La), river (Ri) and wetland (We), aggregate of snow (Sn), soil moisture (SM) and canopy (Cn), and reservoir (Re)

compartments. (b) Water storage change in river, wetland, soil moisture, lake and snow storages. Canopy storage is not included because the cumulated change is in the order of $1 \cdot 10^{-3}$ mm LWH. Estimates are given in millimetres of land water height (mm LWH) and of sea level equivalent (mm SLE).

## 6.    Discussion

### 6.1    TWSA temporal components: seasonality and long-term variability

#### 6.1.1    Linear trend: comparison to independent estimates

In terms of the linear trend of the ensemble mean, Wg_gl overestimates the positive contribution of TWSA to OMC by 30–50% as compared to the GRACE TWSA trends from this study (Tables 3 and 6). However, if we assume 70% deficit irrigation in GWD regions and GPCC precipitation, the simulated trend is within the GRACE uncertainty bounds (Table 3). We consider this variant more likely because 1) GPCC is based on a much larger number of station records than CRU (see

Figure 2 of Schneider et al., 2014) and 2) assuming that farmers in GWD areas have optimal irrigation conditions seems implausible (Döll et al., 2014). Despite this, we included this assumption in the design of the model variants as an upper-bound of GWD (Figure 7). GRACE estimates from other studies (Rietbroek et al., 2016; Reager et al., 2016; Blazquez et al., 2018) suggest much smaller continental water mass losses to oceans (Table 6). Differences between GRACE-based TWSA trends from this study and from independent sources are of the same order of magnitude than differences between GRACE-





and model-based TWSA trends from this study. This suggests that GRACE-based TWSA trends are very sensitive to the multiple processing parameters applied to the GRACE Level-2 data (Blazquez et al., 2018).

**Table 6:** Comparison between trends of TWSA, LGWSA, LWSA, LWSA$_{hum}$ and LWSA$_{clim}$ from this study and literature. TWSA trends from this study were derived from GRACE observations and from integrated WaterGAP (Wg_gl) in anthropogenic mode. LWSA trends were obtained by subtracting LGWSA trends from TWSA trends based on either GRACE or Wg_gl. LWSA$_{clim}$ trends were obtained by
subtracting LWSA$_{hum}$ based on Wg_gl from LWSA based on either GRACE or Wg_gl. Positive trends translate to ocean mass gain, whereas negative trends translate to ocean mass loss. Estimates are given in millimetres of sea level equivalent per year (mm SLE yr$^{-1}$).

| Study | Method | Time period | TWSA | LGWSA | LWSA | LWSA$_{hum}$ | LWSA$_{clim}$ |
|---|---|---|---|---|---|---|---|
| Dieng et al. (2017) | I + M | Jan 1993–Dec 2015 | 1.00 (0.91[c]) | 0.76 ± 0.08 (0.62 ± 0.03[c]) | 0.24 ± 0.09 (0.29[c]) | | 0.12 (-0.05[c]) |
| | | Jan 2004–Dec 2015 | 1.03 (0.81 ± 0.20[b]/1.19[c]) | 0.78 ± 0.07 (0.76 ± 0.03[c]) | 0.25 ± 0.08 (0.05[b,c]/0.43[c]) | | |
| Rietbroek et al. (2016) | G | Apr 2002–Jun 2014[a] | 0.09 (0.64 ± 0.20[b]/0.94[c]) | 0.38 ± 0.07 (0.74 ± 0.03[c]) | -0.29 ± 0.26 (-0.10[b,c]/0.20[c]) | | |
| Reager et al. (2016) | G + I | Apr 2002–Nov 2014[a] | 0.32 ± 0.13 (0.64 ± 0.20[b]/0.97[c]) | 0.65 ± 0.09 (0.74 ± 0.03[c]) | -0.33 ± 0.16 (-0.10[b,c]/0.23[c]) | | -0.71 ± 0.20 (-0.41[b,c]/-0.09[c]) |
| Blazquez et al. (2018) | G | Aug 2002–Jun 2014[a] | 0.07 ± 0.12[d] (0.61 ± 0.20[b]/0.93[c]) | | | | |
| Schrama et al. (2014) | G | Jan 2003–Dec 2013 | | 0.44 ± 0.03 (0.75 ± 0.03[c]) | | | |
| Bamber et al. (2018) | I | Sep 2002–Aug 2006[a] | | 0.48 ± 0.09 (0.71 ± 0.03[c]) | | | |
| | | Sep 2007–Aug 2011 | | 0.55 ± 0.08 (0.73 ± 0.03[c]) | | | |
| Zemp et al. (2019) | O | Sep 2006–Aug 2016 | | 0.56 ± 0.04 (0.80 ± 0.03[c]) | | | |
| Dieng et al. (2015) | B | Jan 2003–Dec 2013 | | | 0.30 ± 0.18 (-0.11[b,c]/0.18[c]) | | |
| Wada et al. (2016) | I | Jan 1993–Dec 2010 | | | | 0.12 ± 0.04 (0.31[c]) | |
| IPCC AR5 | I | Jan 1993–Dec 2011 | | | | 0.38 ± 0.12 (0.31[c]) | |

[a] Only three (CSR_rl06sh, ITSG_2018 and JPL_rl06sh) out of four GRACE solutions where considered (GFZ_rl06sh was excluded because of lack of values in 2002). [b] Trends based on GRACE data sets used in this study. [c] Trends based on modelled data sets (Wg_gl) used in this study. [d] Uncertainty estimates in the source paper are expressed in 1.65σ. Here, they are expressed in 1σ. I: multiple
independent estimates; M: modelling; G: GRACE data; O: observations; B: global water mass budget.

The overestimation of the TWSA positive contribution by Wg_gl may arise from uncertainty in both the LWSA and LGWSA components. There is a rather good agreement between the LGWSA trend from GGM, on the one hand, and from Dieng et al. (2017) and Reager et al. (2016), on the other hand. The agreement to Dieng et al. (2017) is not surprising, since their estimates were obtained by averaging three data sets, including a GGM data set used in Marzeion et al. (2015) (update





from Marzeion et al., 2012). Nevertheless, according to the GRACE-based estimates from Rietbroek et al. (2016) and Schrama et al. (2014), GGM overestimates the LGWSA contribution. The more recent non-GRACE-based estimates from Bamber et al. (2018) and Zemp et al. (2019) are in better agreement with the estimates from GGM, even though still too small in comparison (Table 6).

The discrepancy between GRACE- and model-based TWSA trends from this study is also reflected in the LWSA trends. If
LGWSA is subtracted from TWSA of Wg_gl, a mass loss on land is computed, while a (small) mass gain on land results if GRACE-based TWSA is used instead (Table 6). This can be related to the findings of Scanlon et al. (2018), which show that LWSA trends summed over 183 basins worldwide (~63% of global continental area excluding the ice sheets) are positive (mass gain on land) for GRACE but negative (mass loss on land) for models. The LWSA trends from Rietbroek et al. (2016) and Reager et al. (2016) suggest a higher water mass gain on continents than our GRACE-based trends (Table 6), reflecting
the discrepancy in TWSA trends, which is not compensated by smaller glacier mass losses in these two studies (Table 6). On the other hand, it is noteworthy that our model-based LWSA trends are in good agreement with other non-GRACE-based trends, namely the ones from Dieng et al. (2015) and Dieng et al. (2017) (Table 6).

The presumed overestimation of the LWSA positive contribution to OMC by Wg_gl may reflect an overestimation of the LWSA$_\text{hum}$ positive contribution and/or an underestimation of the LWSA$_\text{clim}$ negative contribution. Our LWSA$_\text{hum}$ trend is in
good agreement with the one reported by the IPCC AR5, but overestimated according to the trend of Wada et al. (2016) (Table 6). Wada et al. (2016) argued that the LWSA$_\text{hum}$ positive contribution of the IPCC AR5 is probably overestimated by a factor of 3, and that this is partly due to the fact that the IPCC AR5 assumes that 100% of GWD ends up in the ocean, whereas their study shows that only 80% of GWD ends up in the ocean. We estimate a GWD trend of 0.39 mm SLE yr$^{-1}$ over 2003–2016 (Table 5), which is within the uncertainty bounds of the trend reported by Wada et al. (2016), 0.30 ± 0.10
mm SLE yr$^{-1}$ over 2002–2014, even if slightly higher. Concerning the LWSA$_\text{clim}$ component, Dieng et al. (2017) computed a positive contribution, whereas Wg_gl computed a negative contribution, suggesting differences between models. Moreover, the trend from Reager et al. (2016) suggests that Wg_gl underestimates continental water mass gain due to climate variability; by assuming GRACE-based TWSA, we obtain a LWSA$_\text{clim}$ trend closer to the estimate of Reager et al. (2016) (Table 6).

**6.1.2  Seasonality**

The small discrepancy between GRACE and Wg_gl in terms of TWSA seasonality (Figure 4b) is partly due to differences in seasonal amplitude. For instance, some years (2006, 2009 and 2011) show smaller simulated seasonal amplitude than what is observed by GRACE. Although we did not investigate this matter at regional scale, we speculate that this might be due to a systematic underestimation of seasonal amplitude in tropical basins by WaterGAP, where the seasonal signal is strongest,
resulting from insufficient storage capacity (Scanlon et al., 2019). At global scale, however, underestimation in tropical basins might be compensated by overestimation in other types of basin.



### 6.1.3    Residual: long-term non-linear variability

The most prominent discrepancies between global mean monthly TWSA from GRACE and Wg_gl are observed in the residual signal (Figure 4f), which contains the inter-annual variability. The inter-annual variability comes almost completely from the LWSA component (Figures 4e and 4f), and more specifically from its climate-driven component (LWSA$_{clim}$ in Figure 5b) at global scale. Cazenave (2018) pointed out that this is arguably the most difficult component in the land water budget to quantify. Humphrey et al. (2016) show that inter-annual anomalies in the GRACE signal can be correlated to anomalies in precipitation (positive correlation) and near-surface temperature (negative correlation); we discuss the relation between precipitation anomalies and inter-annual variability in Section 6.3.1. Furthermore, inter-annual fluctuations in LWS can reflect the occurrence of atmospheric circulation patterns, like the El Niño Southern Oscillation (Cazenave and Llovel, 2010; Llovel et al., 2011; Cazenave et al., 2012). The discrepancy between the residual signal in GRACE and Wg_gl is more prominent in some years (Figure 4f). In particular, we can identify the intense La Niña event of 2010/2011 (positive peak indicating a water mass gain on land) and the intense El Niño event of 2015/2016 (negative peak indicating a water mass loss on land). If we rely on the validity of the GRACE time series, despite the significant gaps in the data for both events, then it can be inferred that, even though Wg_gl reproduces the events to some extent, it underestimates their intensity. We discuss further the relation between the El Niño Southern Oscillation (ENSO) occurrence and inter-annual variability in Section 6.3.2.

### 6.2  Limitations of study

Simulated global TWSA is the result of aggregating water storage change estimates corresponding to nine individual water storage compartments and 64432 grid cells. There is uncertainty in each single estimate (due to uncertain climate input, assumptions related to water use, model parameters etc.). However, errors in individual storage compartments and at smaller spatial scales may average out once aggregated at global scale. In this section, we discuss about the limitations of our reconstruction of time series of global TWSA and thus mass transfer from continents to oceans. Limitations in our approach are related to the integration of glacier data as an input to WaterGAP, to the global models (GGM and WaterGAP) used to compute LGWSA and LWSA and to missing components that were not accounted for in this study.

### 6.2.1    Glacier data integration approach

The glacier data integration significantly improved the simulation of the global mean TWSA linear trend by WaterGAP (Figure 4c and 4d). However, this approach does not give appreciably different results from simply adding the separately estimated LGWSA and LWSA components (hereafter the "addition approach") at global scale (Figure 3). According to data used in this study, we estimate that glaciers cover 0.38% of the global continental area (excluding the ice sheets), which is smaller than the estimate of Bamber et al. (2018), amounting to 0.50%. However, the area effectively accounted for by integrated WaterGAP amounts to 0.34% (~11% of global glacier area is neglected, see Section 3.3), resulting in a reduction of its global land area ranging from 0.39% in 1948 to 0.34% in 2016. Thus, it is not strange that the reduction of the land





area had an insignificant effect at global scale. We speculate that, at basin scale, the glacier data integration approach might
show significantly different results from the addition approach.

Moreover, our approach has limitations regarding the fate of the internally calculated glacier runoff. One of the sources of uncertainty related to the contribution of glaciers to SLR is the interception of glacier runoff by land; it is still vastly assumed that glacier runoff flows directly to the ocean, with no delay or interception by water storage compartments (Church et al., 2013). In our approach, we assume that glacier runoff is intercepted by surface storages, but not by sub-surface (soil
and groundwater) storages. We made this assumption because we have no means of assessing how much glacier runoff is intercepted by sub-surface storages at global scale.

### 6.2.2    Global modelling of LGWSA

Previous studies (Marzeion et al., 2015; Slangen et al., 2017) have shown agreement between GGM and other global glacier models. For instance, according to Marzeion et al. (2015), the reconstruction of global glacier mass change during the
twentieth century by GGM is consistent with the ones obtained from other methods of reconstruction (see their Figure 1). However, note that this might simply mean that the methods are consistently wrong. In addition, using an extrapolation of glaciological and geodetic observations, Zemp et al. (2019) estimate that glaciers (outside the ice sheets) contributed $23 \pm 14$ mm SLE to global-mean SLR from 1961 to 2016. We estimate a contribution of 25 mm SLE with GGM over the same period, which is remarkably consistent with the estimate from Zemp et al. (2019). Furthermore, our evaluation of GGM
performance (Section 4.1) shows that this model can reproduce well the observed mean seasonality of winter accumulation and summer ablation; this is not always the case for global glacier models (Fig.4, Hirabayashi et al., 2010).

Despite the fact that we consider GGM estimates to be state-of-the-art, they are subject to multiple sources of uncertainty. Input data (climate forcing and glacier outlines), simplification of physics in the model, observation data used for calibration and the calibration itself are among the main sources. GGM includes uncertainty estimates related to annual glacier mass
change time series. However, we did not include these uncertainty estimates in our assessment (we only included the trend uncertainty, which corresponds to a $1\sigma$ standard uncertainty) for consistency reasons (i.e. most data sets used for the assessment have unknown uncertainties).

### 6.2.3    Global modelling of LWSA

Uncertainty in WaterGAP estimates is related to both the $LWSA_{hum}$ and $LWSA_{clim}$ components. Modelling of GWD, which
is both related to climate variability and human water use, is of key importance, as global water storage trends computed with WaterGAP are particularly sensitive to these variations (Müller Schmied et al., 2014). Global GWD is highly linked to irrigation groundwater abstraction. The estimation of gross and net irrigation groundwater abstraction is not a trivial task, as it relies mainly on statistical data and assumptions, and depends on climate input (Döll et al., 2016). In the past, the rate of global and regional GWD has been subject to much debate (Döll et al., 2014; Wada et al., 2017). According to Wada et al.





(2017), most studies likely overestimated the cumulative contribution of GWD to global SLR during the twentieth and early twenty-first century. Our GWD estimates are very likely overestimated under optimal irrigation (Döll et al., 2014).

Modelling reservoir storage and operation is also subject to multiple sources of uncertainty (e.g. quality of reservoir data base, algorithms and assumptions used in model). In the present study, we did not include model variants differing from one another in the way reservoirs are handled. Wada et al. (2017) estimated a global reservoir storage capacity of 7968 km³ (~22

mm SLE) until 2014. WaterGAP has a global reservoir storage capacity of 5764 km³ (~16 mm SLE), as it only simulates the largest 1082 reservoirs (Section 2.1.1). Furthermore, by assuming that on average 85% of the reservoir capacity is used and taking into account seepage (i.e. adding additional water that seeps underground), Wada et al. (2017) estimated a potential total water impoundment in reservoirs of ~29 mm SLE. Upon application of the reservoir operation algorithm implemented in WaterGAP, we estimate an actual total water impoundment of ~10 mm SLE, which corresponds to roughly 63% of the

global reservoir capacity. Our estimate of total water impoundment might be underestimated because WaterGAP only simulates the largest reservoirs and because it does not account for seepage. On the other hand, Wada et al. (2017) might overestimate the additional water due to seepage, as well as the fraction of the design capacity that is in reality filled (85% according to their assumption).

$LWSA_{clim}$ is largely affected by uncertain climate input data. As stated by Döll et al. (2016), this remains one of the main

challenges in the development and application of GHMs. Precipitation and radiation data have been identified as strong drivers of water storage change (Müller Schmied et al., 2014; Müller Schmied et al., 2016; Humphrey et al., 2016). Our assessment accounts for part of the uncertainty related to precipitation input data by considering two different climate forcings (WFDEI-GPCC and WFDEI-CRU). We believe that WFDEI-GPCC is likely to be more reliable than WFDEI-CRU because 1) the monthly time series of gridded precipitation from GPCC used to bias-adjust WFDEI-GPCC are based on

more observation stations (Müller Schmied et al., 2016) and 2) GRACE-derived trends of TWSA in 186 large river basins correlate much more with trends computed by WaterGAP if GPCC precipitation is used (Scanlon et al., 2018). Despite this, both forcings are not well suited for trend analysis as a consequence of the bias correction, which significantly affects trends of climatic variables such as temperature and precipitation (Hempel et al., 2013; Weedon et al., 2014). TWSA trends simulated by WaterGAP are most likely affected by this caveat regarding the climate forcing. Moreover, note that, given the

complex interactions and feedbacks in the climate system, we could not, unlike for $LWSA_{hum}$, isolate the different components of $LWSA_{clim}$. Nevertheless, we discuss about some of its main drivers in Section 6.3.

### 6.2.4    Missing components

The Caspian Sea (largest endorheic lake worldwide), which was one of the largest contributors to global lake water storage loss during the twentieth century (Milly et al., 2010), is missing from our assessment because the model grid, based on the

WATCH-CRU land–sea mask, does not include it. The contribution of this endorheic lake to SLR has been estimated to 0.109 ± 0.004 mm SLE yr⁻¹ during the period 2002–2014 (Wada et al., 2017) and 0.114 ± 0.013 mm SLE yr⁻¹ during the



period April 2002 to March 2016 (Wang et al., 2018). Moreover, WaterGAP does not account for land cover change. This means that the impact of anthropogenic-induced phenomena such as deforestation is neglected. Wada et al. (2017) estimated that net global deforestation, through runoff increase and water release from oxidation and plant storage, contributed ~0.035

mm SLE yr$^{-1}$ to SLR over 2002–2014.

### 6.3  Analysis of climate-driven LWSA

### 6.3.1    Correlation to precipitation

We found a correlation of r = 0.63 between GPCC precipitation anomalies and LWSA$_{clim}$ based on GPCC precipitation, and of r = 0.72 with CRU precipitation (Figure 8). Furthermore, we found a correlation of r = 0.87 by comparing the difference

between the two precipitation time series to the difference between the two LWSA$_{clim}$ time series. From these results, we deduce that precipitation is most likely the main driver of LWSA$_{clim}$ at global scale. By comparing global precipitation anomalies from CRU TS3.10 and GPCC v5, Harris et al. (2014) identified a correlation of r = 0.88 over 1951–2009 (see their Table II and Figure 10). We identified a correlation of r = 0.86 over 1948–2016 between the precipitation time series used in this study. This is noteworthy, given the fact that GPCC is based on a much larger number of station records than

CRU. Note that the high correlation between the two data sets means that our ensemble underestimates the uncertainty in global TWSA due to precipitation input data.

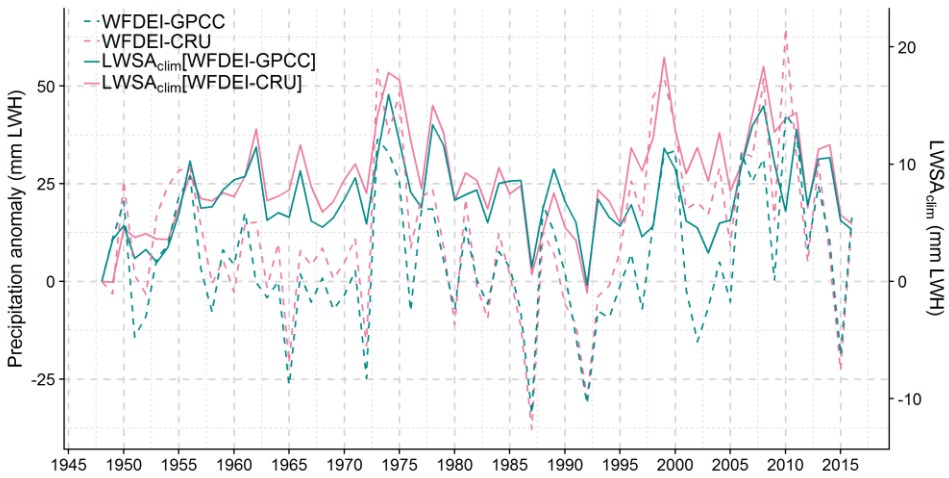

**Figure 8:** Correlation between global annual climate-driven LWSA and precipitation anomaly, 1948 to 2016. Precipitation (rainfall plus snowfall) anomalies correspond to the WFDEI-GPCC and WFDEI-CRU forcings used in this study (Section 2.1.1). Climate-driven LWSA

(LWSA$_{clim}$[WFDEI-GPCC] and LWSA$_{clim}$[WFDEI-CRU]) were obtained with integrated WaterGAP in naturalized mode (see Table 2). Anomalies are relative to the year 1948 and given in millimetres of land water height (mm LWH).



### 6.3.2    Relation to ENSO

Previous studies have shown that LWSA$_{clim}$ is affected by internal multi-year climate variability such as ENSO (Llovel et al., 2011; Cazenave et al., 2012; Boening et al., 2012). We looked at the relation between the residual signal (i.e. non-linear inter-annual variability) in TWSA (Figure 9a) and short-term natural climate variability related to ENSO and expressed as Multivariate ENSO Index (MEI) version 2 intensities (Wolter and Timlin, 1993; Wolter and Timlin, 1998) over 1980–2016 (Figure 9b). The period was chosen according to the availability of MEI data. The MEI, which combines both oceanic and

atmospheric variables, is an indicator of ENSO intensity. Note that, during El Niño phases, TWS tends to decrease; this is due to a rainfall deficit over the continents (mostly the tropics). During La Niña phases, the opposite is observed. Based on MEI intensities, we identified four major La Niña events (MEI < 1) and five major El Niño events (MEI > 1) during 1980 to 2016. According to our results, part of the signature in simulated TWSA inter-annual variability reflects ENSO-driven climate variability. We can observe a continental water storage decrease during El Niño phases, as opposed to a continental

water storage increase during La Niña phases. Differences due to precipitation input data are significant. The impact of the La Niña event of 1988/1989 is more prominent with GPCC precipitation; this is related to higher precipitation anomalies (i.e. wetter conditions) with GPCC (Figure 8). The opposite is observed during the La Niña event of 1998/2000, and can be explained in the same way. Moreover, note that there is no difference between the two irrigation variants, because LWSA$_{hum}$ mainly affects long-term linear variability (Figures 5b and 6). By studying the GRACE record at regional scale, Wang et al.

(2018) showed that the sensitivity to ENSO modulations is more prominent in the global exorheic (i.e. draining into the ocean) system, as opposed to the global endorheic (i.e. landlocked) system (see their Figure 2). Within the exorheic system, tropical basins (particularly the Amazon) are more sensitive to these modulations (Llovel et al., 2011).



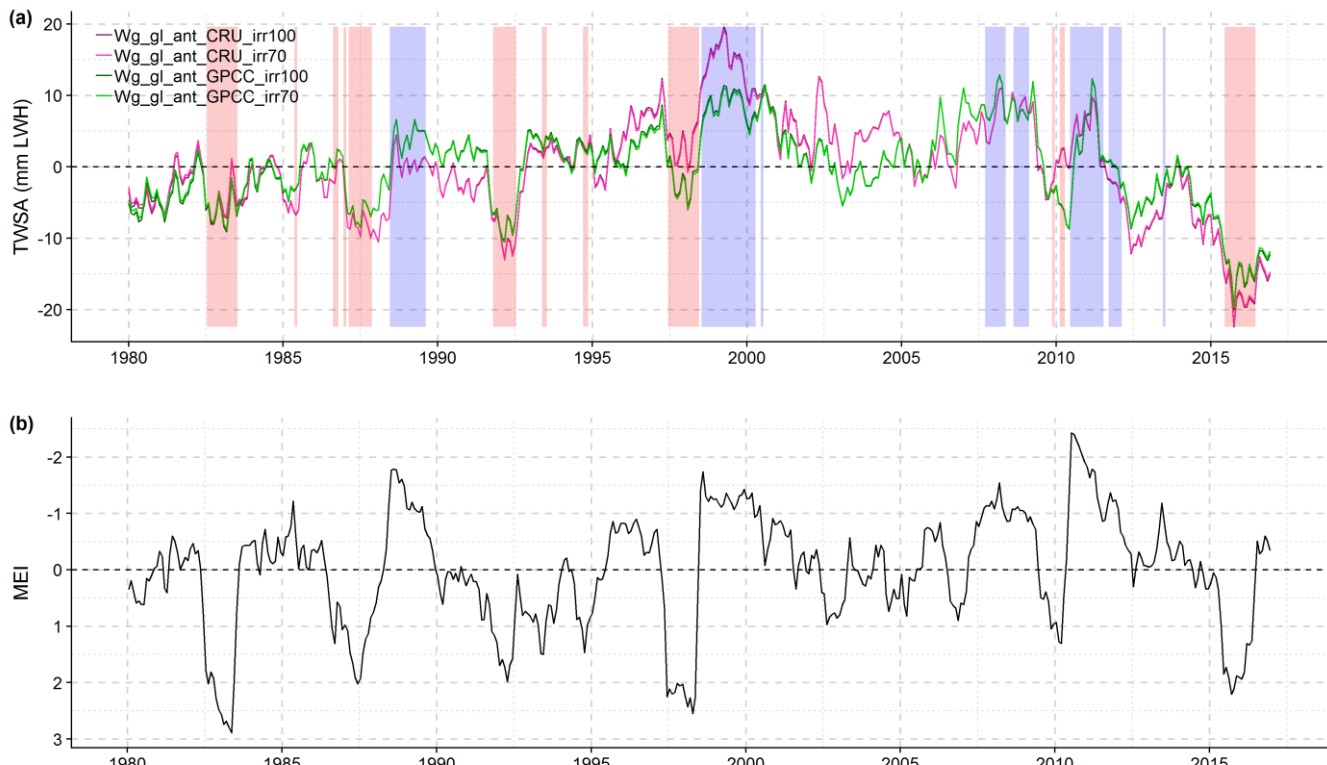

**Figure 9:** Relation between global monthly TWSA and ENSO, January 1980 to December 2016. (a) Residual TWSA (non-linear inter-annual variability) computed with four variants of integrated WaterGAP (Wg_gl) in anthropogenic mode (Table 1). Anomalies are relative to the mean over January 1990 to December 2010 and given in millimetres of land water height (mm LWH). (b) Intensities in Multivariate ENSO Index (MEI) version 2 (produced by NOAA). Positive MEI values indicate El Niño and negative values indicate La Niña phases. El Niño (red) and La Niña (blue) phases are highlighted in plot (a) whenever the MEI gets larger than 1 or smaller than -1, respectively. Note the reversed vertical axis in (b).

## 7. Conclusions

In order to quantify water transfers between continents and oceans over the period 1948 to 2016, we integrated the output of the state-of-the-art global glacier model GGM into the state-of-the-art global hydrological model WaterGAP. This resulted in a non-standard version of WaterGAP that is able to simulate the impact of glacier mass changes and runoff on continental water flows and storages in a consistent manner, and thus accounts for the variations in all continental water storage compartments. In order to take into account major hydrological modelling uncertainties, the integrated WaterGAP model was run under different assumptions of irrigation water use and with different precipitation input data sets. Time series of global mean monthly TWSA simulated with this ensemble were evaluated by comparing them to estimates from an ensemble of GRACE solutions for the time period January 2003 to August 2016. A remarkable agreement between observed and modelled global mean monthly TWSA time series was found, with a high agreement with respect to seasonality and a



likely small overestimation of the water storage decline. We found no significant differences for global mean TWSA between our integrated modelling approach and simply adding the independently obtained LGWSA and LWSA components.

According to our model-based reconstruction, we conclude that continental water mass loss resulted in an ocean mass gain equivalent to 34–41 mm SLE during 1948–2016. Continents (including glaciers) lost water at an accelerated rate over time, with a contribution to OMC of 0.18 mm SLE yr$^{-1}$ over 1948–1975, 0.58 mm SLE yr$^{-1}$ over 1976–2002 and 1.18 mm SLE yr$^{-1}$

over 2003–2016 (Table 5). Global glacier mass loss accounted for 81% of the cumulated mass loss over 1948–2016, while the remaining 19% was lost from other continental water storage compartments (LWS). Changes in LWS over 1948–2016 were dominated by the impact of direct human interventions, namely water abstractions and impoundment of water in reservoirs. LWS mass loss from water abstractions (15–21 mm SLE), which showed an acceleration over time, offset LWS mass gain from reservoir water impoundment (8 mm SLE), which showed a deceleration over time. GWD (13–19 mm SLE)

is strongly linked to water abstractions for irrigation purposes. Climate-driven variability in LWS is highly correlated to precipitation variations and is also influenced by multi-year modulations related to ENSO.

Significant uncertainty in our assessment arises from the simulation of human-driven LWS changes. Modelling of GWD, which is highly sensitive to irrigation water use assumptions, is particularly challenging. Furthermore, simulated climate-driven LWS changes are affected by uncertainty in the climate input data. Despite the limitations of our model-based

approach and the remaining challenges, our assessment gives interesting insights on the main individual mass components and drivers of global water transfers from continents to oceans, as well as on possible routes for model improvement. More research is required to better constrain the simulation of human water use in GHMs. Finally, future research should go beyond the global scale by identifying the main regions contributing to water transfers between continents and oceans.

**Data availability**

All GRACE-based and model-based data sets used in this study are available upon request from the corresponding author. The glacier SMB observational data used in this study are publicly available and be downloaded from the following link; https://wgms.ch/products_ref_glaciers/ (accessed on April 18$^{th}$ 2018). The MEI data used in this study are publicly available and can be downloaded from the following link; https://www.esrl.noaa.gov/psd/enso/mei/data/meiv2.data (accessed on July

10$^{th}$ 2019).

**Author contribution**

PD and DC designed the study. DC conducted background research, implemented computer code for the integrated WaterGAP model version, conducted model simulations, prepared the WaterGAP data, performed the formal analysis and drafted the initial manuscript with substantial revisions from PD. BG, BM, HMS and PD discussed the results and edited the

initial manuscript. BG prepared the GRACE data, provided a script for the computation of linear trends and the temporal



disaggregation of TWSA time series and contributed to the analysis of GRACE-based TWSA. PD contributed to the analysis of model-based TWSA and individual components. BM provided GGM simulation data and contributed to the analysis of glacier mass changes. HMS supported the implementation of computer code and provided a script for the aggregation of model-based TWSA over the global continental area. JM prepared the gridded glacier-related data.

**Competing interests**

The authors declare that they have no conflict of interest.

**Acknowledgements**

We thank Tim Trautmann from the Institute of Physical Geography of the Goethe University Frankfurt for his valuable comments to improve the first draft of the paper. We thank the investigators of the World Glacier Monitoring Service 760 network as well as the NOAA Earth System Research Laboratory's Physical Sciences Division for free and open access to their data sets. This study was enabled by support from the European Space Agency (ESA) through its Sea-Level Budget Closure CCI project (4000119910/17/I-NB).

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
