# Peer review of "Assessing global water mass transfers from continents to oceans over the period 1948–2016"

_Hydrology and Earth System Sciences, 2019_

## Referee Comment (RC1) · Anonymous Referee #1 · 7 Mar 2020

Caceres et al introduce a new version of the WaterGAP global hydrological model that is able to simulate total continental water storage anomalies (TWSA) by integrating a global glacier model (Marzeion et al., 2012). They reach important calculations of continental water storage anomalies and of their implications for ocean mass change. They also quantify (again, and new estimates) TWSA due to groundwater mass abstractions and impoundment in reservoirs. These calculations are worth publication, due to their urgent need with rising sea water levels and decreasing terrestrial water availability. Although I am aware of the big effort beyond this study (an accumulated effort with the ongoing improvement of WaterGap), it is not ready for publication and I recommend a major revision.

The knowledge gap or research questions addressed by this manuscript are not clear.

[Figure]

I also think that the manuscript is too long, giving too much information on all models and related details, and limited in the discussion and comparison with other studies. Also, the authors need to put their results in the context of the urgency to understand TWSA and OMC that precisely requires these results. I find also some issues with structure, that again, break the main thread and hide the really important results.

First, the question is what is the real aim of the manuscript? I think that the authors should state more clearly the research question or hypothesis driving this study, than just focusing it as a new version or integration of WaterGAP. Which is the real aim of the study: 1) to quantify (or rather update or give other versions of) glacier and non-glacier contributions to sea level, with more emphasis on glacier or non-glacier?, 2) to validate TWSA from WaterGap with GRACE, to update WaterGap with the glacier module integration? It is not clear at all. I recommend a main focus, with appropriate redistributed weight across the manuscript.

The introduction should be restructured to really focus on the research question addressed, the identification of knowledge gaps and objectives specifically the knowledge gaps. Glaciers, water use, sea, ocean mass change. I know that combining all of these concepts together and reducing the state of the art to an introduction or discussion is not easy, but I think it is worth the effort. There are many efforts attempting to do similar objectives, with completely different methods, that should be mentioned (for glacier melt contributions, impoundment and withdrawals of water, etc). If GRACE is not the main objective, I suggest reducing considerably the emphasis on it in the introduction. Also, missing important references, see next paragraph.

And again, the discussion, a comparison with other works is necessary, and to put the results of the study in context. Some of the references I here mention could help, as many others. Discussion is completely missing. For instance, can these estimates be compared in some way with others? Anomalies related to water storage and/or consumption by irrigation and reservoir impoundment (Chao et al., 2008; Hoekstra and Mekonnen, 2012; Jaramillo and Destouni, 2015; Stefanie Rost et al., 2008), glacier

contributions to oceans (Braithwaite and Raper, 2002; Giesen and Oerlemans, 2013; Huss and Hock, 2015; Jacob et al., 2012; Meier, 1984; Radić and Hock, 2014).

I have a feeling that Section 6.2 does not belong there, and it is also containing much information that it is not important and dilutes the main message of the manuscript, from my perspective. But again, I may be wrong, depending on the main aim of the article. Can much of it be moved to Supplementary information. Instead, a good discussion of the results and comparison with other studies could fill that gap.

The methods section is very hard to read, or in other words, very hard to focus while reading it. I assume it starts in L. 120 and finishes in L. 330? This should be explicit. There are so many acronyms, too many, I would just use the most important ones. For example what is the purpose of OMC, it is not a variable. The information on models and data sets is important and should be generally included in a methods section, but in this case, due to the massive amount of information due tot eh complexity of the study, I suggest just leaving the most important methods and sending the details to supplementary information. Also, because now the red thread is completely lost by the end of the methods, and since the objectives are not very clear in the introduction.

The results. . .where do they start by the way, in Model evaluation? Also, are they focused on the comparison with GRACE? That is why I ask if that is the main aim of the article.

For figure 4 and 5, can I recommend an additional simple barplot figure showing the TWSA with and without the accounting of glacier melt, with uncertainty ranges?

Another suggestion, a brief explanation somewhere how the term "anomaly" on land and change of mass in the ocean are related.

The conclusions should also pinpoint the main objectives of the introduction, and focus in what is really important.

The y-axis of Fig. 4 and 5 mean different things but have the same level or no level at

all. The level is missing, only units, and complicates the understanding of the results.

L. 466 The word "contribution" is not appropriate here, I would delete it. In general, the use of the word "contribution" is very subjective, can you be more direct in its real meaning. What is a "contribution to ocean mass change" really, increase or decrease in ocean mass? Why is it an addition to the continents?

Just mentioning that the result of Figure 6 is very interesting, illustrative and explains many questions that I had before regarding Global mean annual human-driven LWSA.

On the other hand, Table 5 is too complicated due to the amount of numbers and acronyms and the lack of explanations, maybe a Figure could be more illustrative? Many of the components have not been introduced before. Same for Figure 7.

With losses, do you mean just negative anomalies?

L. 22-25 What do these results have to do with the main aim of the article?

L. 31 – Missing reference (Chao, 2008)

L. 34-35 I don't see the purpose of this sentence.

I think there are too many acronyms, I would only use the most important ones.

L138-140 So why are you using them then?

References

Braithwaite, R.J., Raper, S.C.B., 2002. Glaciers and their contribution to sea level change. Phys. Chem. Earth Parts ABC 27, 1445–1454. https://doi.org/10.1016/S1474-7065(02)00089-X

Chao, B.F., Wu, Y.H., Li, Y.S., 2008. Impact of Artificial Reservoir Water Impoundment on Global Sea Level. Science 320, 212–214. https://doi.org/10.1126/science.1154580

Giesen, R.H., Oerlemans, J., 2013. Climate-model induced differences in the 21st century global and regional glacier contributions to sea-level rise. Clim. Dyn. 41,

3283–3300. https://doi.org/10.1007/s00382-013-1743-7

Jacob, T., Wahr, J., Pfeffer, W.T., Swenson, S., 2012. Recent contributions of glaciers and ice caps to sea level rise. Nature 482, 514–518. https://doi.org/10.1038/nature10847

Jaramillo, F., Destouni, G., 2015. Local flow regulation and irrigation raise global human water consumption and footprint. Science 350, 1248–1251. https://doi.org/10.1126/science.aad1010 Meier, M.F., 1984. Contribution of Small Glaciers to Global Sea Level. Science 226, 1418–1421. https://doi.org/10.1126/science.226.4681.1418

Radić, V., Hock, R., 2014. Glaciers in the Earth's Hydrological Cycle: Assessments of Glacier Mass and Runoff Changes on Global and Regional Scales. Surv. Geophys. 35, 813–837. https://doi.org/10.1007/s10712-013-9262-y

Rost, Stefanie, Gerten, D., Bondeau, A., Lucht, W., Rohwer, J., Schaphoff, S., 2008. Agricultural green and blue water consumption and its influence on the global water system. Water Resour. Res. 44, W09405. https://doi.org/10.1029/2007WR006331

Rost, S., Gerten, D., Heyder, U., 2008. Human alterations of the terrestrial water cycle through land management. Adv Geosci 18, 43–50. https://doi.org/10.5194/adgeo-18-43-2008
* * *

---

## Referee Comment (RC2) · Anonymous Referee #2 · 27 Mar 2020

This manuscript by Cáceres et al. applied an updated version of WaterGAP hydrological model, which integrates the global glacier model GGM, to simulate the contributions of different continental water storage components, excluding ice sheets in Greenland and Antarctica and their peripheral glaciers, to the ocean mass change from 1948 to 2016. Multiple model variants, with different forcing (precipitation) inputs and irrigation scenarios, and with inclusion/exclusion of various human influences, were applied in order to constrain the model uncertainty and separate anthropogenic impacts from climate impacts. The modeled total water storage anomalies (in terms of inter-annual trend, seasonality, and residuals) were validated against an ensemble of GRACE spherical harmonics solutions, and the modeled anomalies of each water component were validated/cross-compared with available observations and/or existing

literature. They concluded that the modeled total water storage (TWS) seasonality agrees remarkably well with GRACE observations, but the modeled TWS inter-annual decline was overestimated. Since their modeled glacier mass trend agrees well with observations, this overestimation mostly stems from other land water storage components, largely due to the challenge of simulating human water abstraction. Consistent with other studies, their simulation shows that the glacier mass loss is the largest contributor (81%) to ocean mass change, and the human-driven hydrology (including water abstraction and reservoir impoundment) contributes the other 19%. Climate-driven variability in land water storage has a very marginal impact on the long-term ocean mass trend.

Overall, I enjoyed reading this paper. Although very lengthy, it is logically structured and well written. The methods are well elaborated, with lots of technical details that help readers understand and potentially replicate their work. The model validations are overall thorough and in-depth, with extensive cross-comparisons with existing literature. So potentially, I find this paper will be another important contribution towards the improved closure of global water budget.

However, I do have some major concerns/comments, mostly about model validation and result implication. I would like to see the authors' responses before a possible acceptance.

My first concern is the selection of GRACE solutions. Obviously, GRACE solutions are important here as they were used as validation benchmarks. The authors applied the ensemble of four spherical harmonics (SH) solutions. I agree with the authors that the derived water storage trend can be sensitive to different GRACE solutions. Therefore, I wonder why the authors opted to only use SH solutions but not mascon solutions. Mascon solutions have merits in improved resolution and signal isolation. I am wondering what kind of influence the inclusion of mascon solutions will have on the model validation (I know some of the mascon solutions result in a smaller global trend, meaning the modeled LWSA trend even more overestimated). I would like to see more

justifications, or at least a little discussion, about the selection of GRACE solutions.

Second, as the authors specified, the Caspian Sea is not included in WaterGAP, so the modeled LWSA excludes the impact of the level decline in the Caspian Sea (which can be substantial). The GRACE solutions, however, include the Caspian Sea in the land mask. I am unclear, at least given what the authors described, whether the GRACE TWSA used for model validation excludes the Caspian Sea or not. If not, the comparison wouldn't be apple-to-apple (GRACE TWSA with the Caspian Sea impact and modeled TWSA without), which leads to an even greater overestimation in the modeled water storage trend.

Third, the modeled impact of reservoir water impoundment may be underestimated, particularly in the recent couple of decades. This is because WaterGAP incorporated the initial version of GRanD (version 1.1), rather than the updated version 1.3 that added another 458 large reservoirs mostly constructed after the year 2000. As a result, the modeled reservoir impact was underestimated after 2000. This underestimation was seen in Figure 6, where LWSA_res flattens and declines during the recent decades. In addition, as the authors mentioned, GRanD only includes the largest reservoirs. Medium-sized and smaller reservoirs, such as partially documented in ICOLD, may add up to a substantial impoundment volume that was not taken into account in the model. These limitations (leading to underestimated LWSA_res) ended up overestimating the declining trend in net human impacts (LWSA_human). The authors may want to discuss more about these limitations about modeling reservoir impacts, and how they affect the conclusions.

A couple of minor comments:

Line 558: "closer to the estimate of Reager et al. (2016)". I found this sentence farfetching as the estimates of Reager et al. (-0.71 mm per year) and of the authors (-0.41 mm per year) have a difference of a factor of 2.

Lines 660-661: This sentence can be misleading. 0.109 mm per year from 2002 to

2014 as reported in Wada et al. (2017), includes the contribution from both the surface water in the Caspian Sea and its influenced groundwater. 0.114 mm per year from 2002 to 2016 as reported in Wang et al. (2018), is the contribution of the entire Caspian Sea Basin, not the Caspian Sea alone. Based on Wang et al. (2018), the contribution of the Caspian Sea alone, i.e., the trend in its surface water volume, is 0.071 +/- 0.006 mm per year (or 25.52+/-2.12 Gt per year). For improved clarity, I recommend that the authors modify this sentence to be something like: "The contribution of this endorheic lake to SLR was estimated to 0.109 +/- 0.004 mm SLE per year (including variations in both surface water and the influenced groundwater) during the period 2002-2014 (Wada et al., 2017) and 0.071 +/- 0.006 mm SLE per year (including only surface water variation) during the period April 2002 to March 2016 (Wang et al., 2018)."

---

## Author Comment (AC1) · 15 May 2020

We thank referee #1 for the very pertinent comments and questions, as well as the positive feedback. We have listed the referee comments and respective replies below. Each set of referee comment (RC) and corresponding author comment (AC) is identified by a number.

1.

RC: The knowledge gap or research questions addressed by this manuscript are not clear. [. . .] First, the question is what is the real aim of the manuscript? I think that the authors should state more clearly the research question or hypothesis driving this study, than just focusing it as a new version or integration of WaterGAP. Which is

the real aim of the study: 1) to quantify (or rather update or give other versions of) glacier and non-glacier contributions to sea level, with more emphasis on glacier or non-glacier?, 2) to validate TWSA from WaterGap with GRACE, to update WaterGap with the glacier module integration? It is not clear at all. I recommend a main focus, with appropriate redistributed weight across the manuscript. [. . .] The introduction should be restructured to really focus on the research question addressed, the identification of knowledge gaps and objectives specifically the knowledge gaps. Glaciers, water use, sea, ocean mass change. I know that combining all of these concepts together and reducing the state of the art to an introduction or discussion is not easy, but I think it is worth the effort. There are many efforts attempting to do similar objectives, with completely different methods, that should be mentioned (for glacier melt contributions, impoundment and withdrawals of water, etc). If GRACE is not the main objective, I suggest reducing considerably the emphasis on it in the introduction. Also, missing important references [. . .].

AC: The aim of the study was stated in L. 104–105:

"Our assessment aimed at quantifying this contribution during the period 1948–2016, as well as identifying its main drivers and components."

However, we agree on the fact that the knowledge gaps, aims and research questions of the study need to be expressed in a more structured way and formulated more clearly. Therefore, we will conscientiously re-formulate the introduction, making sure that all of these elements are clearly addressed and that the connection between them is apparent. Moreover, we will reduce the emphasis on the comparison to GRACE, since this is not the main focus of the study. With respect to the research questions, they will be formulated in the form of a list as follows:

"Through this comprehensive assessment, we aim to address the following questions:

1. How did changes of total water storage on the continents of the Earth (except Greenland and Antarctica) contribute to ocean mass changes (and thus sea-level rise)

during the period 1948–2016? (section 3.2.1)

2. Which continental storages underwent the most significant mass changes during this period? (sections 3.2.1 and 3.2.4)

3. How have man-made reservoirs and human water abstractions affected water storage on the continents? (sections 3.2.1 and 3.2.2)

4. What were the main climatic drivers of glacier-free land water storage changes? (section 3.2.3)

5. To what extent can we rely on our modelling approach to quantify global-scale water storage changes on the continents? (sections 3.1 and 4.2)"

Note that the order of the research questions intends to convey that the long-term assessment of TWSA and individual components (research questions 1, 2, 3 and 4) is the main focus of the manuscript, whereas the validation of the applied modelling approach (research question 5) is a secondary focus that is indispensable, as only based on this validation can the reader judge the reliability of our assessment. The fact that the main focus is disaggregated into four research questions shows that this main focus is broad and points towards an exhaustive assessment of TWSA and individual components. Note also that the section number(s) indicated in parentheses for each question corresponds to the new structure of the results and discussion sections (see AC 4).

2.

RC: I also think that the manuscript is too long, giving too much information on all models and related details, and limited in the discussion and comparison with other studies.

AC: We generally agree on the fact that the manuscript is too long. Therefore, we will reduce to some extent the information on models, data sets and methods as described in AC 5. Furthermore, Table 5 will be shortened (see Table 1 in the supplement of

Interactive
comment
the response) by moving part of the results to the supplementary information, and the introduction and conclusions will be shortened as far as possible. On the other hand, we disagree with the suggestion that our manuscript is limited in the discussion and comparison with other studies for the reasons stated in AC 3.

3.

RC: Also, the authors need to put their results in the context of the urgency to understand TWSA and OMC that precisely requires these results. [. . .] And again, the discussion, a comparison with other works is necessary, and to put the results of the study in context. Some of the references I here mention could help, as many others. Discussion is completely missing. For instance, can these estimates be compared in some way with others? Anomalies related to water storage and/or consumption by irrigation and reservoir impoundment (Chao et al., 2008; Hoekstra and Mekonnen, 2012; Jaramillo and Destouni, 2015; Stefanie Rost et al., 2008), glacier contributions to oceans (Braithwaite and Raper, 2002; Giesen and Oerlemans, 2013; Huss and Hock, 2015; Jacob et al., 2012; Meier, 1984; Radić and Hock, 2014).

AC: We have rather extensively discussed our results in the light of other works. For instance, in section 6.1.1, we have compared our linear trends of TWSA, LGWSA, LWSA, LWSA_hum and LWSA_clim to estimates from multiple recently published studies (see Table 6 in the manuscript). Furthermore, section 6.2 also contains numerous references to other studies. In section 6.2.2, global glacier contributions to oceans are discussed (1) by referring to studies (Marzeion et al., 2015; Slangen et al., 2017) in which the results of the glacier model used in our study have already been compared to other models, including the ones from Radić et al. (2014) and Huss and Hock (2015), (2) by comparing our estimate to the one from Zemp et al. (2019), which provides the state-of-the-art observational baseline of global glacier contributions to oceans, and (3) by referring to the results of Hirabayashi et al. (2010), who evaluated the mean seasonality computed by their model in a comparable way to ours. In section 6.2.3, we have compared our estimate of the contribution of global reservoir impoundment to oceans

to the most comprehensive assessment done up to now (Wada et al., 2017), which is based on and updates the results of Chao et al. (2008). As to groundwater storage anomalies related to irrigation groundwater abstraction, we have discussed this in relation to other studies in a qualitative way only; we recognize that some quantitative comparison to other estimates is missing here.

Some of the references that you mention can be of interest for qualitative discussion. Rost et al. (2008) quantified the impact of anthropogenic land cover change on the global terrestrial water balance by applying a dynamic global vegetation and water balance model (LPJmL). Their conclusions can be of interest to discuss the implications of lacking this influence in our assessment (section 6.2.4, "Missing components") due to the fact that WaterGAP does not include land cover change. However, many other references are not very relevant or suitable for comparison in our opinion. For instance, the estimates reported by Jaramillo and Destouni (2015) are relative to 100 large basins which represent 35% of the global land area excluding Antarctica, and thus could not be compared to our global-scale estimates. Moreover, given the fact that the manuscript is already very long, we do not intend to include too many further references.

In the revised manuscript, we will compare our groundwater depletion estimates to other works and will include a reference to the study of Rost et al. (2008). As to the need of putting our results more in the context of urgency, this will be resolved by re-formulating the knowledge gaps in the introduction (see AC 1).

4.

RC: I find also some issues with structure, that again, break the main thread and hide the really important results. [. . .] I have a feeling that Section 6.2 does not belong there, and it is also containing much information that it is not important and dilutes the main message of the manuscript, from my perspective. But again, I may be wrong, depending on the main aim of the article. Can much of it be moved to Supplemen-

tary information. Instead, a good discussion of the results and comparison with other studies could fill that gap.

AC: We consider that the content of section 6.3 (we assume that the referee comment refers to section 6.3 and not 6.2) should be left as part of the manuscript, rather than moved to the supplementary information. As mentioned in AC 1, our assessment of TWSA and individual components is intended to be exhaustive. Even if we did not disaggregate LWSA_clim into individual contributions (as opposed to LWSA_hum, which was disaggregated into LWSA_res and LWSA_abs) due to technical limitations of the model and general complexity, we still intended to investigate the main drivers behind this component. This will be reflected by research question 4 (see AC 1). This being said, we agree that the placement of this section in terms of structure is not ideal, and thus will move it from the discussion to the results (new section 3.2.3, see below). Also, we will make the content of this section more concise. The structure of the results and discussion sections will be modified as follows:

(3) Results

(3.1) Model evaluation

(3.1.1) Comparison of observed and simulated annual and seasonal glacier mass changes

(3.1.2) Comparison of observed and simulated global mean TWSA during the period January 2003 to August 2016

(3.2) Global water transfer from continents to oceans over the period 1948–2016

(3.2.1) Contributions of glaciers (LGWSA), climate-driven land water storage (LWSA_clim) and human-driven land water storage (LWSA_hum)

(3.2.2) Contributions of reservoirs (LWSA_res) and water abstractions (LWSA_abs)

(3.2.3) Relation between climate and land water storage

(3.2.4) Contributions of individual water storage compartments

(4) Discussion

(4.1) TWSA temporal components: seasonality and long-term variability

(4.1.1) Linear trend: comparison to independent estimates

(4.1.2) Seasonality

(4.1.3) Residual: long-term non-linear variability

(4.2) Limitations of study

(4.2.1) Glacier data integration approach

(4.2.2) Global modelling of LGWSA

(4.2.3) Global modelling of LWSA

(4.2.4) Missing components

5.

RC: The methods section is very hard to read, or in other words, very hard to focus while reading it. I assume it starts in L. 120 and finishes in L. 330? This should be explicit. [...] The information on models and data sets is important and should be generally included in a methods section, but in this case, due to the massive amount of information due to the complexity of the study, I suggest just leaving the most important methods and sending the details to supplementary information. Also, because now the red thread is completely lost by the end of the methods, and since the objectives are not very clear in the introduction.

AC: We agree that the information on models and data sets should be part of the methods section. Therefore, we will merge sections 2 and 3, "Models and data" and "Methods", respectively, into one section entitled "Models, data and methods". The structure of this section will then be:

(2) Models, data and methods

(2.1) Models

(2.1.1) Global hydrological model (WaterGAP)

(2.1.2) Global glacier model (GGM)

(2.2) Data

(2.2.1) Modelled land water storage change (LWSA)

(2.2.2) Modelled land glacier water storage change (LGWSA)

(2.2.3) Glacier mass change from in situ observations

(2.2.4) GRACE-derived total continental water storage change (TWSA)

(2.3) Methods

(2.3.1) Integration of GGM glacier data into WaterGAP

(2.3.2) Model experiments

The content that will be assigned to section 2.2.3 will roughly correspond to the content of the old section 3.1 ("Evaluation of GGM glacier mass change with in situ observations"), but more concise and focused on the observed glacier mass change data set. Section 2.3.2 corresponds to the old section 3.4 ("Overview of WaterGAP variants").

Concerning the amount of information on models, data sets and methods, we are of the opinion that most of it should remain in the main text, because it is necessary to understand the results. We are also encouraged to do so based on the feedback of referee #2, who states that, although the manuscript is very lengthy, the methods are well elaborated, and the technical details help readers understand and potentially replicate our work. Nevertheless, for the sake of reducing the length of the manuscript, we will make the content of section 2.1.1 ("Global hydrological model (WaterGAP)") more concise, we will move the old section 3.2 ("Pre-processing of gridded GGM output

data") to the supplementary information and we will modify Table 1 to make it more compact (see Table 2 in the supplement of the response).

6.

RC: There are so many acronyms, too many, I would just use the most important ones. For example what is the purpose of OMC, it is not a variable.

AC: We agree that the vast number of acronyms makes it difficult for the reader to stay focused. Therefore, we will not use the following acronyms, as they are not considered essential: SLR (sea-level rise), OMC (ocean mass change), SWB (surface water bodies), SMB (surface mass balance) and GWD (groundwater depletion). Furthermore, in paragraphs which contain too many acronyms (e.g. L. 464–471), we will replace part of them with the corresponding long names in order to reduce the load of acronyms. Moreover, in Table 5, we will include both the long name and the corresponding acronym of each variable (see Table 1 in the supplement of the response).

7.

RC: The results. . .where do they start by the way, in Model evaluation? Also, are they focused on the comparison with GRACE? That is why I ask if that is the main aim of the article.

AC: In order to avoid confusions as to where the results begin, we will modify the structure as shown in AC 4.

8.

RC: For figure 4 and 5, can I recommend an additional simple barplot figure showing the TWSA with and without the accounting of glacier melt, with uncertainty ranges?

AC: We are not sure what you meant here. Would the values (bars) represented in the barplot still represent the TWSA time series (i.e. one bar per month/year), or would they represent the linear trends of contribution of TWSA to ocean mass change over

the period considered by the respective figure? In any case, for both Figure 4 and 5, we do not see how adding an additional barplot would help the understanding of the reader.

However, in the case of Figure 5, including a barplot alongside each graph with the corresponding linear trends over 1948–2016 could be beneficial. First of all, the barplot related to the upper graph (TWSA) would replace Table 4 in the manuscript and, in this way, contribute to save some space (see Figure 1 in the supplement of the response). Secondly, the barplot related to the bottom graph (TWSA components) would allow us to include these additional results (see Figure 1 in the supplement of the response).

9.

RC: Another suggestion, a brief explanation somewhere how the term "anomaly" on land and change of mass in the ocean are related.

AC: The following sentence will be added in the introduction:

"An overall positive (negative) water storage anomaly in a given compartment is equivalent to a water mass gain (loss) in this compartment, thus constituting a "negative" ("positive") contribution to ocean mass change."

10.

RC: The conclusions should also pinpoint the main objectives of the introduction, and focus in what is really important.

AC: the conclusions will be re-formulated to be more in accordance with the aims and research questions stated in the revised introduction (AC 1). In addition, the first paragraph, with summarises the modelling approach and its validation, will be shortened.

11.

RC: The y-axis of Fig. 4 and 5 mean different things but have the same level or no level at all. The level is missing, only units, and complicates the understanding of the

results.

AC: At the end of the caption of Fig. 4, it is explained that "Anomalies are relative to the mean over the period January 2006 to December 2015 and given in millimetres of land water height (mm LWH)." The same sentence is found in the caption of Fig. 5, on that the anomaly is relative to the annual value of 1948. We are not completely sure what you meant by "the same level or no level at all", however we would say that the level (i.e. what a value of zero means) is well described by these sentences and the reference levels are different. We think it is better to not have the same reference level in both figures as the reference level in Fig. 4 is due to GRACE data availability only, while Fig. 5 is to show changes since the beginning of the study period.

12.

RC: L. 466 The word "contribution" is not appropriate here, I would delete it. In general, the use of the word "contribution" is very subjective, can you be more direct in its real meaning. What is a "contribution to ocean mass change" really, increase or decrease in ocean mass? Why is it an addition to the continents?

AC: In order to clarify what we mean by negative or positive contribution to ocean mass change, we will add an explanatory sentence in the introduction (see AC 9). In the discussion manuscript, we have distinguished between "contribution to ocean mass change" and "contribution to TWSA". The second expression was used when referring to mass changes of individual components of TWSA; in e.g. section 5.3, the water storage compartments that gain mass (e.g. reservoir) represent positive contributions to TWSA, whereas the ones that lose mass (e.g. glacier) represent negative contributions to TWSA. A positive (negative) contribution to TWSA is nothing more than a water mass gain (loss) in the continents, which translates into a negative (positive) contribution to ocean mass change. To avoid confusions, in the revised manuscript we will only use the word "contribution" in relation to ocean mass change. Regarding TWSA, we will simply refer to water mass gains or losses. Furthermore, we will modify L. 465–468

as follows:

"Glacier mass loss is the dominant component of the TWSA mass budget (Figure 5b), with LGWSA accounting for 81% of the cumulated water mass loss from continents to oceans over 1948–2016. Overall, the contribution of LWSA to ocean mass change, which is dominated by its human-driven component (Figure 5b), is also positive, representing 19% of the cumulated water mass loss from continents."

13.

RC: On the other hand, Table 5 is too complicated due to the amount of numbers and acronyms and the lack of explanations, maybe a Figure could be more illustrative? Many of the components have not been introduced before. Same for Figure 7.

AC: In order to reduce the complexity of Table 5, we will modify it (see Table 1 in the supplement of the response). The variables and corresponding trends that do not appear in the modified Table 5 (as compared to the one in the manuscript) will be moved to an independent table in the supplementary information. On the other hand, we disagree with the suggestion that many of the components in Table 5 and Figure 7 had not been introduced before. The components corresponding to the individual water storage compartments were introduced in section 2.1.2 ("Computation of LWSA").

14.

RC: With losses, do you mean just negative anomalies?

AC: That is correct. Negative water storage anomalies in a given compartment indicate water mass losses in this compartment (see AC 9).

15.

RC: L. 22–25 What do these results have to do with the main aim of the article?

AC: The main aim of the article is to give an exhaustive long-term (1948–2016) assessment of TWSA and individual components. The results that you pointed out give

the contribution of the LWSA_res and LWSA_abs components to ocean mass change over the considered period, while relating the large mass decrease in the groundwater storage compartment to LWSA_abs. In addition, the LWSA_clim component is related to some of its main drivers. In the revised manuscript, these results will correspond to research questions 2, 3 and 4 (see AC 1).

16.

RC: L. 31 – Missing reference (Chao, 2008)

AC: We will add this reference in connection to this line.

17.

RC: L. 34–35 I don't see the purpose of this sentence.

AC: The purpose of this sentence is to introduce the mass component of sea-level change, i.e. ocean mass change. In the revised manuscript, the sentence will be slightly modified to:

"Primarily, sea-level change can be decomposed into a steric component (i.e. thermal expansion and salinity change) and a mass component (i.e. ocean mass change)."

18.

RC: L138–140 So why are you using them then?

AC: The WFDEI climate forcing is not optimal for trend analysis due to varying station density in space and time related to the bias correction of monthly precipitation sums based on observation-based products (GPCC or CRU data). Despite this non-negligible caveat, WFDEI is still considered the state-of-the-art when it comes to large-scale hydrological impact studies. For instance, it has been used to force multiple impact models such as global hydrological models in the framework of the Inter-Sectoral Impact Model Intercomparison Project (https://www.isimip.org).

Adjusting reanalysis as e.g. ERA-Interim (WFDEI is based on this reanalysis data set) by observation-based products such as monthly GPCC or CRU precipitation allows for the consideration of additional data to scale the monthly sums while keeping the temporal daily variability of the reanalysis (Weedon et al., 2011; Weedon et al., 2014). Together with the snow undercatch corrections that are included in some of the state-of-the-art climate forcings (such as WFDEI), this renders hydrological impact studies more plausible. Müller Schmied et al. (2016) show in their Table 4 the effect of precipitation monthly bias correction, e.g. PGFv2.1 incorporating monthly CRU TS3.21, which results in a close match to initial CRU TS3.21 data, but also the effect of snow undercatch correction on the other climate forcings shown there. Kauffeldt et al. (2013) show in their Figure 8 that physically implausible runoff ratios would be obtained based solely on precipitation observations (CRU) in high-latitude or high-altitude areas without the implementation of a snow undercatch correction method. Hence, the inclusion of monthly observation-based precipitation data sets together with snow undercatch corrections improves the usability of such reanalyses.

In the case of GPCC, the data center provides a wide range of products. The typical product that is incorporated in reanalysis for hydrological impact studies (as e.g. WFDEI) is the so-called "Full Data Monthly" product, described as follows: "for the period 1891 to 2016 based on quality-controlled data from all stations in GPCC's data base available at the month of regard with a maximum number of more than 53,000 stations in 1986/1987. This product is optimized for best spatial coverage and use for water budget studies" (https://www.dwd.de/EN/ourservices/gpcc/gpcc.html). It is the most comprehensive data set (in both, space and time) they currently offer for advancing reanalysis at the given spatial resolution of 0.5°. A specifically dedicated product for trend analysis from GPCC is "HOMPRA Europe (V1)", described as follows: "a homogenized and quality-controlled data product suitable for trend analysis. The product is provided for the years 1951–2005 and is based on 5536 stations" (https://www.dwd.de/EN/ourservices/gpcc/gpcc.html). Even though this product is better suited for trend analysis, it is restricted to Europe and to the period 1951–2005, and

thus not useful for most hydrological impact assessments.

To sum up, we justify our choice of climate forcing data sets by a) the lack of global-scale long-term precipitation measurements to generate time series that are specifically suited for trend analysis and b) the raised value of reanalysis when additional precipitation observation data together with snow undercatch corrections are included. We agree that this is not well expressed in the manuscript and thus have modified the corresponding sentences as follows:

[revised manuscript text omitted]

---

## Author Comment (AC2) · 15 May 2020

We thank referee #2 for the very pertinent comments and questions, as well as the positive feedback. We have listed the referee comments and respective replies below. Each set of referee comment (RC) and corresponding author comment (AC) is identified by a number.

1.

RC: My first concern is the selection of GRACE solutions. Obviously, GRACE solutions are important here as they were used as validation benchmarks. The authors applied the ensemble of four spherical harmonics (SH) solutions. I agree with the authors that the derived water storage trend can be sensitive to different GRACE solutions.

Therefore, I wonder why the authors opted to only use SH solutions but not mascon solutions. Mascon solutions have merits in improved resolution and signal isolation. I am wondering what kind of influence the inclusion of mascon solutions will have on the model validation (I know some of the mascon solutions result in a smaller global trend, meaning the modeled LWSA trend even more overestimated). I would like to see more justifications, or at least a little discussion, about the selection of GRACE solutions.

AC: We agree that, in principle, mascons may be beneficial for many applications, especially in basin-scale analysis. However, this is not our use-case. Deploying mascons was initially discussed internally, but subsequently not considered for several reasons (without order):

- Being part of the ESA CCI Sea Level Budget Project (SLBC_cci) let us aim for a complementary GRACE product to ocean mass change with preferably identical corrections and uncertainty assessment procedures, i.e. a degree-60 spherical-harmonics (SH) based approach that includes recent Glacial Isostatic Adjustment corrections after Caron et al. (2018), which is – to our knowledge – not available with mascons. In the meantime, there may be options without individual corrections included (e.g. CSR), but then the entire approach runs the risk of losing consistency.

- Mascons generally use geophysical a-priori constraints / regularization in space and time, which we find too interfering. The implementation of time-correlation as in JPL-RL06M introduces (small) changes to previous months after an update, which affects trend determination and comparability.

- We undertook an in-depth uncertainty assessment for the SH approach, which we understand much better than mascon uncertainties, where provided. Ours includes a combined time-dependent estimation of contributions from low-degree replacements (geo-center, flattening), GIA, leakage and noise, which lets us derive uncertainty ranges for individually picked temporal base-lines, as promoted in SLBC_cci.

- One of the key elements in our GRACE continental solution is an oceanic leakage

buffer: we implemented a dedicated procedure in order to integrate out-leaking signal over the oceanic area and correct for mean ocean change therein. For this, we need to employ a specific latitudinal-dynamic buffer mask (∼300 km) that exactly nestles to the WGHM land-water mask. This is not easily accomplished with the mascons' dedicated masks, also not with JPL's CRI version. Even though mascons are (by design) not as affected by leakage as SH solutions, mascons would still require additional coastal processing due to fractional land-ocean separation. The mean-ocean-mass correction for such cells would require a product dedicated to ocean-mass and not ocean bottom pressure, as commonly provided with some mascon solutions. Adjusting for this effect is possible but, again, adds to inconsistencies in the overall approach.

- If gain factors were to be used, they could not be consistently applied in our global approach, as this also includes combined glacier- and hydrologically affected cells.

We could theoretically derive solutions solely based on mascons for comparison, yes. However, these would not fulfil our requirements in a consistent way. Therefore, we believe it would not be worth the effort for this study. With regard to filtering and statistical noise-cancelling, we should also say that our approach is suitable for the global case, but may be significantly weaker in regional applications.

2.

RC: Second, as the authors specified, the Caspian Sea is not included in WaterGAP, so the modelled LWSA excludes the impact of the level decline in the Caspian Sea (which can be substantial). The GRACE solutions, however, include the Caspian Sea in the land mask. I am unclear, at least given what the authors described, whether the GRACE TWSA used for model validation excludes the Caspian Sea or not. If not, the comparison wouldn't be apple-to-apple (GRACE TWSA with the Caspian Sea impact and modeled TWSA without), which leads to an even greater overestimation in the modelled water storage trend.

AC: Our GRACE continental mass change estimates do indeed include the Caspian

Sea area. In our global approach, it would technically not be possible to treat it consistently as an ocean, primarily for the following reasons: (1) we extend the continental integration kernel onto the ocean area for leakage, but need to subtract the mean ocean change therefrom; which does not connect to the Caspian Sea for obvious reasons. (2) In recent versions of the ocean correction, we restore AOD1b (GAD) background models; which do not exist for the Caspian Sea, because it is not an ocean. And (3) if we wanted to apply the extended leakage-mask technique to the Caspian Sea, the latter would almost entirely be covered by the integration kernel and, hence, still contain the mass change signal. Which, in turn, would have to be corrected for unknown mean 'ocean' (Sea) mass change each month. One might just mask out the Caspian Sea from the integration kernel, but thereby one would also introduce methodological inconsistency. We will discuss internally, whether such an option would be more beneficial than a trend-wise discussion. In either case, we understand that this issue needs more clarification.

3.

RC: Third, the modeled impact of reservoir water impoundment may be underestimated, particularly in the recent couple of decades. This is because WaterGAP incorporated the initial version of GRanD (version 1.1), rather than the updated version 1.3 that added another 458 large reservoirs mostly constructed after the year 2000. As a result, the modeled reservoir impact was underestimated after 2000. This underestimation was seen in Figure 6, where LWSA_res flattens and declines during the recent decades. In addition, as the authors mentioned, GRanD only includes the largest reservoirs. Medium-sized and smaller reservoirs, such as partially documented in ICOLD, may add up to a substantial impoundment volume that was not taken into account in the model. These limitations (leading to underestimated LWSA_res) ended up overestimating the declining trend in net human impacts (LWSA_human). The authors may want to discuss more about these limitations about modeling reservoir impacts, and how they affect the conclusions.

AC: We thank you for this very pertinent comment. We recognize that the modelled impact of reservoir water impoundment is underestimated by the model, particularly during the last decades of the studied period, due to the fact that the additional reservoirs documented in GRanD v1.3, as compared to GRanD v1.1, are not accounted for by the model. There are ongoing efforts to incorporate these reservoirs in WaterGAP. However, this enhancement is still in progress and will only be available upon the release of the next model version. Therefore, as suggested by you, we will discuss about this limitation and how it affects our results and conclusions. The following text will replace L. 640–643 in the discussion:

"Wada et al. (2017) might overestimate the additional water due to seepage, as well as the fraction of the design capacity that is in reality filled (85% according to their assumption). However, the estimate of our study is likely an underestimation of the impoundment of water in man-made reservoirs because WaterGAP only simulates the largest reservoirs and does not account for seepage. In addition, WaterGAP incorporates the reservoirs from the GRanD v1.1 database, but not the additional ones from the new GRanD v1.3 release (http://globaldamwatch.org). GRanD v1.3 includes 458 additional reservoirs as compared to GRanD v1.1. Out of 458 reservoirs, 447 were put in operation between 1948 and 2016. Out of these 447 reservoirs, 173 have a total capacity of at least 0.5 km^3 and thus would be simulated as reservoirs by WaterGAP. The cumulated total capacity of these 173 reservoirs amounts to 599 km^3. The remaining 274 smaller reservoirs have a cumulated total capacity of 62 km^3. Out of the 173 large reservoirs, 164 were put in operation between 2000 and 2016. Taking into account that we computed an actual total water impoundment of roughly 63% of the global reservoir capacity, we can infer that incorporating the additional large reservoirs would lead to an additional impoundment of 378 km^3 (1.05 mm SLE) over 1948–2016, thus increasing total impoundment of water from 8 to 9 mm SLE, i.e. from 22 mm LWH to 25 mm LWH (compare Fig. 6). Most of the additional impoundment not taken into account in this study (369 km^3, 1.02 mm SLE) occurred in the period 2000–2016. Therefore, WaterGAP is expected to overestimate the contribution of water storage

on the continents during the GRACE period by approximately 0.06 mm SLE/yr, which explains part of the overestimation as compared to GRACE (Table 3)."

4.

RC: Line 558: "closer to the estimate of Reager et al. (2016)". I found this sentence far-fetching as the estimates of Reager et al. (-0.71 mm per year) and of the authors (-0.41 mm per year) have a difference of a factor of 2.

AC: This sentence will be modified as follows:

"the trend from Reager et al. (2016) suggests that Wg_gl underestimates continental water mass gain due to climate variability; by assuming GRACE-based TWSA, we obtain a more negative LWSA_clim trend, however still differing from the estimate of Reager et al. (2016) by roughly a factor of 2"

5.

RC: Lines 660–661: This sentence can be misleading. 0.109 mm per year from 2002 to 2014 as reported in Wada et al. (2017), includes the contribution from both the surface water in the Caspian Sea and its influenced groundwater. 0.114 mm per year from 2002 to 2016 as reported in Wang et al. (2018), is the contribution of the entire Caspian Sea Basin, not the Caspian Sea alone. Based on Wang et al. (2018), the contribution of the Caspian Sea alone, i.e., the trend in its surface water volume, is 0.071 +/- 0.006 mm per year (or 25.52+/-2.12 Gt per year). For improved clarity, I recommend that the authors modify this sentence to be something like: "The contribution of this endorheic lake to SLR was estimated to 0.109 +/- 0.004 mm SLE per year (including variations in both surface water and the influenced groundwater) during the period 2002–2014 (Wada et al., 2017) and 0.071 +/- 0.006 mm SLE per year (including only surface water variation) during the period April 2002 to March 2016 (Wang et al., 2018)."

AC: We thank you for having pointed out this misinterpretation of ours. We will modify the sentence as proposed. In addition, we will also cite a trend reported by Milly et al.

(2010); 0.06 mm yr-1 SLE over 1992–2002.

References

Caron, L., Ivins, E. R., Larour, E., Adhikari, S., Nilsson, J., and Blewitt, G.: GIA Model Statistics for GRACE Hydrology, Cryosphere, and Ocean Science, Geophys. Res. Lett., 45, 2203–2212, doi:10.1002/2017GL076644, 2018.

Milly, P. C. D. C., Cazenave, A., Famiglietti, J. S., Gornitz, V., Laval, K., Lettenmaier, D. P., Sahagian, D. L., Wahr, J. M., and Wilson, C. R.: Terrestrial Water-Storage Contributions to Sea-Level Rise and Variability, in: Understanding Sea-level rise and variability, Church, J. A. (Ed.), John Wiley & Sons, Chichester, 226–255, 2010.

Reager, J. T., Gardner, A. S., Famiglietti, J. S., Wiese, D. N., Eicker, A., and Lo, M.-H.: A decade of sea level rise slowed by climate-driven hydrology, Science (New York, N.Y.), 351, 699–703, doi:10.1126/science.aad8386, 2016.

Wada, Y., Reager, J. T., Chao, B. F., Wang, J., Lo, M.-H., Song, C., Li, Y., and Gardner, A. S.: Recent Changes in Land Water Storage and its Contribution to Sea Level Variations, Surv Geophys, 38, 131–152, doi:10.1007/s10712-016-9399-6, 2017.

---

## Author Response (AR1)

**Final author response**

Sections 1 and 2 contain the final author response to the reviews from referee #1 and referee #2, respectively. In each section, each set of referee comment (RC) and corresponding author comment (AC) is identified by a number. Concerning the changes made to the original manuscript by the author (described within the author comments) in response to the referee comments, we refer the reader to the marked-up revised manuscript, which can be found at the end of this document, as well as the non-marked-up revised manuscript and supplementary information. Note that, for each change, we indicate the corresponding page and line numbers. However, note that line numbers are erroneous in the marked-up revised manuscript; this means that line numbers should only be considered in relation to the non-marked-up revised manuscript. Page numbers, on the other hand, are consistent between the marked-up and non-marked-up versions.

**1. Final response to comments from Referee #1**

**1.1**
**RC:** The knowledge gap or research questions addressed by this manuscript are not clear.
[…]
First, the question is what is the real aim of the manuscript? I think that the authors should state more clearly the research question or hypothesis driving this study, than just focusing it as a new version or integration of WaterGAP. Which is the real aim of the study: 1) to quantify (or rather update or give other versions of) glacier and non-glacier contributions to sea level, with more emphasis on glacier or non-glacier?, 2) to validate TWSA from WaterGap with GRACE, to update WaterGap with the glacier module integration? It is not clear at all. I recommend a main focus, with appropriate redistributed weight across the manuscript.
[…]
The introduction should be restructured to really focus on the research question addressed, the identification of knowledge gaps and objectives specifically the knowledge gaps. Glaciers, water use, sea, ocean mass change. I know that combining all of these concepts together and reducing the state of the art to an introduction or discussion is not easy, but I think it is worth the effort. There are many efforts attempting to do similar objectives, with completely different methods, that should be mentioned (for glacier melt contributions, impoundment and withdrawals of water, etc). If GRACE is not the main objective, I suggest reducing considerably the emphasis on it in the introduction. Also, missing important references […].

**AC:** The aim of the study was stated in L. 104–105 of the original manuscript:

*"Our assessment aimed at quantifying this contribution during the period 1948–2016, as well as identifying its main drivers and components."*

However, we agree on the fact that the knowledge gaps, aims and research questions of the study need to be expressed in a more structured way and formulated more clearly. Therefore, we have conscientiously re-formulated the introduction, making sure that all of these elements are clearly addressed and that the connection between them is apparent (L. 30–107 / pg. 1–4). The emphasis on the comparison to GRACE has been reduced, since this is not the main focus of the study. The research questions have been clearly formulated in the form of a list (L. 92–100 / pg. 3–4). Note that the order of the research questions intends to convey that the long-term assessment of TWSA and individual components (research questions 1, 2, 3 and 4) is the main focus of the manuscript, whereas the validation of the applied modelling approach (research question 5) is a secondary focus that is indispensable, as only based on this validation can the reader judge the reliability of our assessment. The fact that the main focus is disaggregated into four research questions shows that this main focus is broad and points towards an exhaustive assessment of TWSA and individual components.

**1.2**

**RC:** I also think that the manuscript is too long, giving too much information on all models and related details, and limited in the discussion and comparison with other studies.

**AC:** We generally agree on the fact that the manuscript is too long. Therefore, we have shortened the introduction (L. 30–107 / pg. 1–4), the content related to models, data sets and methods (L. 108–290 / pg. 4–10) and the content of section 6.3 (Section 3.2.4 in revised manuscript / L. 449–479 / pg. 18–20). Table 1 has been rendered more compact through the usage of placeholders (pg. 9) and Table 5 (Table 4 in revised manuscript / pg. 18) has been shortened by moving part of the results to the supplementary information (Table S3 in the supplementary information / pg. 4). On the other hand, we disagree with the suggestion that our manuscript is limited in the discussion and comparison with other studies for the reasons stated in AC 1.3.

**1.3**

**RC:** Also, the authors need to put their results in the context of the urgency to understand TWSA and OMC that precisely requires these results.
[…]
And again, the discussion, a comparison with other works is necessary, and to put the results of the study in context. Some of the references I here mention could help, as many others. Discussion is completely missing. For instance, can these estimates be compared in some way with others? Anomalies related to water storage and/or consumption by irrigation and reservoir impoundment (Chao et al., 2008; Hoekstra and Mekonnen, 2012; Jaramillo and Destouni, 2015; Stefanie Rost et al., 2008), glacier contributions to oceans (Braithwaite and Raper, 2002; Giesen and Oerlemans, 2013; Huss and Hock, 2015; Jacob et al., 2012; Meier, 1984; Radić and Hock, 2014).

**AC:** We have rather extensively discussed our results in the light of other works. For instance, in section 6.1.1 (Section 4.11 in revised manuscript), we have compared our linear trends of TWSA, LGWSA, LWSA, LWSA$_{hum}$ and LWSA$_{clim}$ to estimates from multiple recently published studies (see Table 5 in revised manuscript). Furthermore, section 6.2 (Section 4.2 in revised manuscript) also contains numerous references to other studies.
In section 6.2.2 (Section 4.2.2 in revised manuscript), global glacier contributions to oceans are discussed (1) by referring to studies (Marzeion et al., 2015; Slangen et al., 2017) in which the results of the glacier model used in our study have already been compared to other models, including the ones from Radić et al. (2014) and Huss and Hock (2015), (2) by comparing our estimate to the one from Zemp et al. (2019), which provides the state-of-the-art observational baseline of global glacier contributions to oceans, and (3) by referring to the results of Hirabayashi et al. (2010), who evaluated the mean seasonality computed by their model in a comparable way to ours.
In section 6.2.3 (Section 4.2.3 in revised manuscript), we have compared our estimate of the contribution of global reservoir impoundment to oceans to the most comprehensive assessment done up to now (Wada et al., 2017), which is based on and updates the results of Chao et al. (2008). As to groundwater storage anomalies related to irrigation groundwater abstraction, we have discussed this in relation to other studies in a qualitative way only; we recognize that some quantitative comparison to other estimates is missing here. Thus, we have introduced a brief comparison between our estimate of groundwater depletion under 70% deficit irrigation over 2003–2016 to the estimate of van Dijk et al. (2014) over a similar period (L. 625–627 / pg. 25–26).

Some of the references that you mention can be of interest for qualitative discussion. Rost et al. (2008) quantified the impact of anthropogenic land cover change on the global terrestrial water balance by applying a dynamic global vegetation and water balance model (LPJmL). Their conclusions are of interest for the discussion about the implications of lacking this influence in our assessment due to the fact that WaterGAP does not include land cover change. Thus, we have introduced a reference to their study (L. 684–686 / pg. 27).
However, many other references are not very relevant or suitable for comparison in our opinion. For instance, the estimates reported by Jaramillo and Destouni (2015) are relative to 100 large basins which represent 35% of the global land area excluding Antarctica, and thus could not be compared to

our global-scale estimates. Moreover, given the fact that the manuscript is already very long, we prefer not to include too many further references.

As to the need of putting our results more in the context of urgency, this has been resolved by re-formulating the knowledge gaps in the introduction.

**1.4**

**RC:** I find also some issues with structure, that again, break the main thread and hide the really important results.

[…]

I have a feeling that Section 6.2 does not belong there, and it is also containing much information that it is not important and dilutes the main message of the manuscript, from my perspective. But again, I may be wrong, depending on the main aim of the article. Can much of it be moved to Supplementary information. Instead, a good discussion of the results and comparison with other studies could fill that gap.

**AC:** We consider that the content of section 6.3 (we assume that the referee comment refers to section 6.3 and not 6.2) should be left as part of the manuscript, rather than moved to the supplementary information. As mentioned in AC 1.1, our assessment of TWSA and individual components is intended to be exhaustive. Even if we did not disaggregate $LWSA_{clim}$ into individual contributions (as opposed to $LWSA_{hum}$, which was disaggregated into $LWSA_{res}$ and $LWSA_{abs}$) due to technical limitations of the model and general complexity, we still intended to investigate the main drivers behind this component. This is now reflected by research question 4 (see AC 1.1). This being said, we agree that the placement of this section in terms of structure is not ideal, and thus have moved it from the discussion to the results (Section 3.2.4 in revised manuscript / L. 449–479 / pg. 18–20). Also, we have made the content of this section more concise. In general, the structure of the results (Section 3 in revised manuscript / L. 295–496 / pg. 11–21) and discussion (Section 4 in revised manuscript / L. 497–686 / pg. 21–27) sections has been reworked.

**1.5**

**RC:** The methods section is very hard to read, or in other words, very hard to focus while reading it. I assume it starts in L. 120 and finishes in L. 330? This should be explicit.

[…]

The information on models and data sets is important and should be generally included in a methods section, but in this case, due to the massive amount of information due to the complexity of the study, I suggest just leaving the most important methods and sending the details to supplementary information. Also, because now the red thread is completely lost by the end of the methods, and since the objectives are not very clear in the introduction.

**AC:** We agree that the information on models and data sets should be part of the methods section. Therefore, we have merged sections 2 and 3, "Models and data" and "Methods", respectively, into one section entitled "Models, data and methods" (Section 2 in revised manuscript / L. 108–290 / pg. 4–10). Concerning the amount of information on models, data sets and methods, we are of the opinion that most of it should remain in the main text, because it is necessary to understand the results. We are also encouraged to do so based on the feedback of referee #2, who states that, although the manuscript is very lengthy, the methods are well elaborated, and the technical details help readers understand and potentially replicate our work. Nevertheless, we have reduced the content as far as possible and have moved section 3.2 ("Pre-processing of gridded GGM output data") to the supplementary information (Section S1.1 in supplementary information / pg. 1).

**1.6**

**RC:** There are so many acronyms, too many, I would just use the most important ones. For example what is the purpose of OMC, it is not a variable.

**AC:** We agree that the vast number of acronyms makes it difficult for the reader to stay focused. Therefore, we have eliminated the use of the following acronyms, as they are not considered essential:

SLR (sea-level rise), OMC (ocean mass change), SWB (surface water bodies), SMB (surface mass balance) and GWD (groundwater depletion). Moreover, in Table 5 (Table 4 in revised manuscript / pg. 18), we have included both the long name and the corresponding acronym of each variable.

**1.7**
**RC:** The results…where do they start by the way, in Model evaluation? Also, are they focused on the comparison with GRACE? That is why I ask if that is the main aim of the article.

**AC:** In order to avoid confusions as to where the results begin, we have modified the structure by merging sections 4 and 5, "Model evaluation" and "Global water transfer from continents to oceans over the period 1948–2016", respectively, into one common section entitled "Results" (Section 3 in revised manuscript / L. 295–496 / pg. 11–21).

**1.8**
**RC:** For figure 4 and 5, can I recommend an additional simple barplot figure showing the TWSA with and without the accounting of glacier melt, with uncertainty ranges?

**AC:** We are not sure what you meant here. Would the values (bars) represented in the barplot still represent the TWSA time series (i.e. one bar per month/year), or would they represent the linear trends of contribution of TWSA to ocean mass change over the period considered by the respective figure? In any case, for both Figure 4 and 5, we do not see how adding an additional barplot would help the understanding of the reader.

However, in the case of Figure 5, including a barplot alongside each graph with the corresponding linear trends over 1948–2016 would be beneficial. First of all, the barplot related to the upper graph (TWSA) would replace Table 4 in the manuscript and, in this way, contribute to save some space. Secondly, the barplot related to the bottom graph (TWSA components) would allow us to include these additional results. Thus, we have modified the figure accordingly (Figure 5 in revised manuscript / pg. 16).

**1.9**
**RC:** Another suggestion, a brief explanation somewhere how the term "anomaly" on land and change of mass in the ocean are related.

**AC:** We have introduced a sentence in the introduction to clarify the meaning of the term "anomaly" (L. 50–51 / pg. 2). Furthermore, we consider that the link between mass anomaly on land and change of mass in the ocean has already been made clear by, for example, having the following sentence in the caption of Table 3 (pg. 15):

*"Negative trends (mass loss) over the continents, expressed in millimetres of land water height (mm LWH, relative to the global continental area without the ice sheets 132.3.106 km$^2$), translate to positive trends (mass gain) over the oceans, expressed in millimetres of sea level equivalent (mm SLE, relative to the global ocean area 361.0.106 km$^2$)."*

**1.10**
**RC:** The conclusions should also pinpoint the main objectives of the introduction, and focus in what is really important.

**AC:** We have performed several modifications to the text (L. 688–716 / pg. 27–28). The main objectives, results and implications are clearly stated.

**1.11**
**RC:** The y-axis of Fig. 4 and 5 mean different things but have the same level or no level at all. The level is missing, only units, and complicates the understanding of the results.

**AC:** At the end of the caption of Fig. 4, it is explained that "Anomalies are relative to the mean over the period January 2006 to December 2015 and given in millimetres of land water height (mm LWH)." The same sentence is found in the caption of Fig. 5, on that the anomaly is relative to the annual value of 1948. We are not completely sure what you meant by "the same level or no level at all", however we would say that the level (i.e. what a value of zero means) is well described by these sentences and the reference levels are different. We think it is better to not have the same reference level in both figures as the reference level in Fig. 4 is due to GRACE data availability only, while Fig. 5 is to show changes since the beginning of the study period.

**1.12**

**RC:** L. 466 The word "contribution" is not appropriate here, I would delete it. In general, the use of the word "contribution" is very subjective, can you be more direct in its real meaning. What is a "contribution to ocean mass change" really, increase or decrease in ocean mass? Why is it an addition to the continents?

**AC:** We agree on the fact that the use of the term "contribution" throughout the manuscript can lead to some misunderstandings. This is because, in the original manuscript, we have used this term in relation to ocean mass change ("contribution to ocean mass change") and to continental water storage change ("contribution to TWSA"). The expression "contribution to TWSA" was used when referring to mass changes of individual components of TWSA; in e.g. section 5.3 (Section 3.2.5 in revised manuscript), the water storage compartments that gain mass (e.g. reservoir) represent positive contributions to TWSA, whereas the ones that lose mass (e.g. glacier) represent negative contributions to TWSA. A positive (negative) contribution to TWSA is nothing more than a water mass gain (loss) in the continents, which translates into a negative (positive) contribution to ocean mass change.

To avoid confusions, in the revised manuscript we have only used the term "contribution" in relation to ocean mass change. Regarding TWSA, we simply refer to water mass gains or losses. The sentence mentioned in the RC has been modified accordingly (L. 423–424 / pg. 17).

**1.13**

**RC:** On the other hand, Table 5 is too complicated due to the amount of numbers and acronyms and the lack of explanations, maybe a Figure could be more illustrative? Many of the components have not been introduced before. Same for Figure 7.

**AC:** In order to reduce the complexity of Table 5 (Table 4 in revised manuscript / pg. 18), we have shortened it by moving part of the results to the supplementary information (Table S3 in supplementary information / pg. 4) and included both the long name and the corresponding acronym of each variable.
On the other hand, we disagree with the suggestion that many of the components in Table 5 (Table 4 in revised manuscript) and Figure 7 (Figure 9 in revised manuscript) had not been introduced before. The components corresponding to the individual water storage compartments had already been introduced in section 2.1.2 (Section 2.2.1 in revised manuscript).

**1.14**

**RC:** With losses, do you mean just negative anomalies?

**AC:** That is correct. Negative water storage anomalies in a given compartment indicate water mass losses in this compartment.

**1.15**

**RC:** L. 22–25 What do these results have to do with the main aim of the article?

**AC:** The main aim of the article is to give an exhaustive long-term (1948–2016) assessment of TWSA and individual components. The results that you pointed out give the contribution of the $LWSA_{res}$ and $LWSA_{abs}$ components to ocean mass change over the considered period, while relating the large mass

decrease in the groundwater storage compartment to LWSA$_{abs}$. In addition, the LWSA$_{clim}$ component is related to some of its main drivers. In the revised manuscript, these results answer research questions 2, 3 and 4 (see AC 1.1).

**1.16**
**RC:** L. 31 – Missing reference (Chao, 2008)

**AC:** We have included the proposed reference in relation to the mentioned sentence (L. 33 / pg. 2).

**1.17**
**RC:** L. 34–35 I don't see the purpose of this sentence.

**AC:** The purpose of this sentence is to introduce the mass component of sea-level change, i.e. ocean mass change. For the revised manuscript, the sentence has been slightly modified (L. 34–36 / pg. 2).

**1.18**
**RC:** L138–140 So why are you using them then?

**AC:** The WFDEI climate forcing is not optimal for trend analysis due to varying station density in space and time related to the bias correction of monthly precipitation sums based on observation-based products (GPCC or CRU data). Despite this non-negligible caveat, WFDEI is still considered the state-of-the-art when it comes to large-scale hydrological impact studies. For instance, it has been used to force multiple impact models such as global hydrological models in the framework of the Inter-Sectoral Impact Model Intercomparison Project (https://www.isimip.org).

Adjusting reanalysis as e.g. ERA-Interim (WFDEI is based on this reanalysis data set) by observation-based products such as monthly GPCC or CRU precipitation allows for the consideration of additional data to scale the monthly sums while keeping the temporal daily variability of the reanalysis (Weedon et al., 2011; Weedon et al., 2014). Together with the snow undercatch corrections that are included in some of the state-of-the-art climate forcings (such as WFDEI), this renders hydrological impact studies more plausible. Müller Schmied et al. (2016) show in their Table 4 the effect of precipitation monthly bias correction, e.g. PGFv2.1 incorporating monthly CRU TS3.21, which results in a close match to initial CRU TS3.21 data, but also the effect of snow undercatch correction on the other climate forcings shown there. Kauffeldt et al. (2013) show in their Figure 8 that physically implausible runoff ratios would be obtained based solely on precipitation observations (CRU) in high-latitude or high-altitude areas without the implementation of a snow undercatch correction method. Hence, the inclusion of monthly observation-based precipitation data sets together with snow undercatch corrections improves the usability of such reanalyses.

In the case of GPCC, the data center provides a wide range of products. The typical product that is incorporated in reanalysis for hydrological impact studies (as e.g. WFDEI) is the so-called "Full Data Monthly" product, described as follows: "for the period 1891 to 2016 based on quality-controlled data from all stations in GPCC's data base available at the month of regard with a maximum number of more than 53,000 stations in 1986/1987. This product is optimized for best spatial coverage and use for water budget studies" (https://www.dwd.de/EN/ourservices/gpcc/gpcc.html). It is the most comprehensive data set (in both, space and time) they currently offer for advancing reanalysis at the given spatial resolution of 0.5°. A specifically dedicated product for trend analysis from GPCC is "HOMPRA Europe (V1)", described as follows: "a homogenized and quality-controlled data product suitable for trend analysis. The product is provided for the years 1951–2005 and is based on 5536 stations" (https://www.dwd.de/EN/ourservices/gpcc/gpcc.html). Even though this product is better suited for trend analysis, it is restricted to Europe and to the period 1951–2005, and thus not useful for most hydrological impact assessments.

To sum up, we justify our choice of climate forcing data sets by a) the lack of global-scale long-term precipitation measurements to generate time series that are specifically suited for trend analysis and b)

the raised value of reanalysis when additional precipitation observation data together with snow undercatch corrections are included. We agree that this is not well expressed in the manuscript and thus have added a sentence to clarify this (L. 124–127 / pg. 4).

**2. Final response to comments from Referee #2**

**2.1**

**RC:** My first concern is the selection of GRACE solutions. Obviously, GRACE solutions are important here as they were used as validation benchmarks. The authors applied the ensemble of four spherical harmonics (SH) solutions. I agree with the authors that the derived water storage trend can be sensitive to different GRACE solutions. Therefore, I wonder why the authors opted to only use SH solutions but not mascon solutions. Mascon solutions have merits in improved resolution and signal isolation. I am wondering what kind of influence the inclusion of mascon solutions will have on the model validation (I know some of the mascon solutions result in a smaller global trend, meaning the modeled LWSA trend even more overestimated). I would like to see more justifications, or at least a little discussion, about the selection of GRACE solutions.

**AC:** We agree that, in principle, mascons may be beneficial for many applications, especially in basin-scale analysis. However, this is not our use-case. Deploying mascons was initially discussed internally, but subsequently not considered for several reasons (without order):

- Being part of the ESA CCI Sea Level Budget Project (SLBC_cci) let us aim for a complementary GRACE product to ocean mass change with preferably identical corrections and uncertainty assessment procedures, i.e. a degree-60 spherical-harmonics (SH) based approach that includes recent Glacial Isostatic Adjustment corrections after Caron et al. (2018), which is – to our knowledge – not available with mascons. In the meantime, there may be options without individual corrections included (e.g. CSR), but then the entire approach runs the risk of losing consistency.
- Mascons generally use geophysical a-priori constraints / regularization in space and time, which we find too interfering. The implementation of time-correlation as in JPL-RL06M introduces (small) changes to previous months after an update, which affects trend determination and comparability.
- We undertook an in-depth uncertainty assessment for the SH approach, which we understand much better than mascon uncertainties, where provided. Ours includes a combined time-dependent estimation of contributions from low-degree replacements (geo-center, flattening), GIA, leakage and noise, which lets us derive uncertainty ranges for individually picked temporal base-lines, as promoted in SLBC_cci.
- One of the key elements in our GRACE continental solution is an oceanic leakage buffer: we implemented a dedicated procedure in order to integrate out-leaking signal over the oceanic area and correct for mean ocean change therein. For this, we need to employ a specific latitudinal-dynamic buffer mask (~300 km) that exactly nestles to the WGHM land-water mask. This is not easily accomplished with the mascons' dedicated masks, also not with JPL's CRI version. Even though mascons are (by design) not as affected by leakage as SH solutions, mascons would still require additional coastal processing due to fractional land-ocean separation. The mean-ocean-mass correction for such cells would require a product dedicated to ocean-mass and not ocean bottom pressure, as commonly provided with some mascon solutions. Adjusting for this effect is possible but, again, adds to inconsistencies in the overall approach.
- If gain factors were to be used, they could not be consistently applied in our global approach, as this also includes combined glacier- and hydrologically affected cells.

We could theoretically derive solutions solely based on mascons for comparison, yes. However, these would not fulfil our requirements in a consistent way. Therefore, we believe it would not be worth the

effort for this study. With regard to filtering and statistical noise-cancelling, we should also say that our approach is suitable for the global case, but may be significantly weaker in regional applications.

**2.2**
**RC:** Second, as the authors specified, the Caspian Sea is not included in WaterGAP, so the modelled LWSA excludes the impact of the level decline in the Caspian Sea (which can be substantial). The GRACE solutions, however, include the Caspian Sea in the land mask. I am unclear, at least given what the authors described, whether the GRACE TWSA used for model validation excludes the Caspian Sea or not. If not, the comparison wouldn't be apple-to-apple (GRACE TWSA with the Caspian Sea impact and modeled TWSA without), which leads to an even greater overestimation in the modelled water storage trend.

**AC:** Our GRACE continental mass change estimates do indeed include the Caspian Sea area. In our global approach, it would technically not be possible to treat it consistently as an ocean, primarily for the following reasons: (1) we extend the continental integration kernel onto the ocean area for leakage, but need to subtract the mean ocean change therefrom; which does not connect to the Caspian Sea for obvious reasons. (2) In recent versions of the ocean correction, we restore AOD1b (GAD) background models; which do not exist for the Caspian Sea, because it is not an ocean. And (3) if we wanted to apply the extended leakage-mask technique to the Caspian Sea, the latter would almost entirely be covered by the integration kernel and, hence, still contain the mass change signal. Which, in turn, would have to be corrected for unknown mean 'ocean' (Sea) mass change each month.

In order to address this issue, we have introduced a sentence that clarifies the fact that the GRACE-based solutions do include Caspian Sea mass changes (L. 676–677 / pg. 27) and performed a first-order evaluation of the possible trend difference between modelled TWSA and GRACE-based estimates by analysing the lake surface integration kernel (L. 677–679 / pg. 27). A brief description of how this first-order evaluation was performed, including two figures, has been incorporated in the supplementary information (Section S4 in supplementary information / pg. 4–6).

**2.3**
**RC:** Third, the modeled impact of reservoir water impoundment may be underestimated, particularly in the recent couple of decades. This is because WaterGAP incorporated the initial version of GRanD (version 1.1), rather than the updated version 1.3 that added another 458 large reservoirs mostly constructed after the year 2000. As a result, the modeled reservoir impact was underestimated after 2000. This underestimation was seen in Figure 6, where LWSA_res flattens and declines during the recent decades. In addition, as the authors mentioned, GRanD only includes the largest reservoirs. Medium-sized and smaller reservoirs, such as partially documented in ICOLD, may add up to a substantial impoundment volume that was not taken into account in the model. These limitations (leading to underestimated LWSA_res) ended up overestimating the declining trend in net human impacts (LWSA_human). The authors may want to discuss more about these limitations about modeling reservoir impacts, and how they affect the conclusions.

**AC:** We thank you for this very pertinent comment. We recognize that the modelled impact of reservoir water impoundment is underestimated by the model, particularly during the last decades of the studied period, due to the fact that the additional reservoirs documented in GRanD v1.3, as compared to GRanD v1.1, are not accounted for by the model. There are ongoing efforts to incorporate these reservoirs in WaterGAP. However, this enhancement is still in progress and will only be available upon the release of the next model version. Therefore, as suggested by you, we have copiously discussed about this limitation and roughly estimated the possible trend difference that we would see if the additional reservoirs in GRanD v1.3 were taken into account in the model (L. 637–651 / pg. 26).

**2.4**

**RC:** Line 558: "closer to the estimate of Reager et al. (2016)". I found this sentence far-fetching as the estimates of Reager et al. (-0.71 mm per year) and of the authors (-0.41 mm per year) have a difference of a factor of 2.

**AC:** We agree with this comment and have modified the mentioned sentence accordingly (L. 548–551 / pg. 23).

**2.5**

**RC:** Lines 660–661: This sentence can be misleading. 0.109 mm per year from 2002 to 2014 as reported in Wada et al. (2017), includes the contribution from both the surface water in the Caspian Sea and its influenced groundwater. 0.114 mm per year from 2002 to 2016 as reported in Wang et al. (2018), is the contribution of the entire Caspian Sea Basin, not the Caspian Sea alone. Based on Wang et al. (2018), the contribution of the Caspian Sea alone, i.e., the trend in its surface water volume, is 0.071 +/- 0.006 mm per year (or 25.52+/-2.12 Gt per year). For improved clarity, I recommend that the authors modify this sentence to be something like: "The contribution of this endorheic lake to SLR was estimated to 0.109 +/- 0.004 mm SLE per year (including variations in both surface water and the influenced groundwater) during the period 2002–2014 (Wada et al., 2017) and 0.071 +/- 0.006 mm SLE per year (including only surface water variation) during the period April 2002 to March 2016 (Wang et al., 2018)."

**AC:** We thank you for having pointed out this misinterpretation of ours. We have modified the sentence as proposed and included a trend reported by Milly et al. (2010) (0.06 mm yr$^{-1}$ SLE over 1992–2002) as well (L. 672–676 / pg. 27).

[revised manuscript text omitted]